# CO$_2$ hydrogenation over Fe-Co bimetallic catalysts with tunable selectivity through a graphene fencing approach

Jiaming Liang[1,4], Jiangtao Liu[2,4], Lisheng Guo [3] ✉, Wenhang Wang[1], Chengwei Wang[1], Weizhe Gao[1], Xiaoyu Guo[1], Yingluo He [1], Guohui Yang [1], Shuhei Yasuda [1] ✉, Bing Liang [2] ✉ & Noritatsu Tsubaki [1] ✉

Tuning CO$_2$ hydrogenation product distribution to obtain high-selectivity target products is of great significance. However, due to the imprecise regulation of chain propagation and hydrogenation reactions, the oriented synthesis of a single product is challenging. Herein, we report an approach to controlling multiple sites with graphene fence engineering that enables direct conversion of CO$_2$/H$_2$ mixtures into different types of hydrocarbons. Fe-Co active sites on the graphene fence surface present 50.1% light olefin selectivity, while the spatial Fe-Co nanoparticles separated by graphene fences achieve liquefied petroleum gas of 43.6%. With the assistance of graphene fences, iron carbides and metallic cobalt can efficiently regulate C-C coupling and olefin secondary hydrogenation reactions to achieve product-selective switching between light olefins and liquefied petroleum gas. Furthermore, it also creates a precedent for CO$_2$ direct hydrogenation to liquefied petroleum gas via a Fischer-Tropsch pathway with the highest space-time yields compared to other reported composite catalysts.

The combustion of fossil fuels for industrialization and transportation, which accompanies the rapid growth of cities, has resulted in the release of massive volumes of CO$_2$ gas into the atmosphere[1]. However, the excessive emissions of CO$_2$ will result in global warming, ocean acidification, and other environmental issues[2,3]. Therefore, the question of dealing with the CO$_2$ emitted during industrial manufacturing has become an urgent concern[4]. Meanwhile, the fabrication of affordable, reliable, and sustainable chemicals as one of the sustainable development goals has attracted increasing attention[5]. A feasible and promising solution for long-term sustainable development is the highly selective catalysis of CO$_2$ into valuable chemicals[6–16], such as light olefins and liquefied petroleum (LPG)[17–20]. As important intermediates in the manufacture of organic products, light olefins are one of the most productive chemicals all over the world, with an amount

exceeding 250 million tons per year[21–23]. Meanwhile, with the worldwide population continuously increasing, the LPG production amount is boosted every year. It is estimated that annual worldwide LPG production will reach 350 million metric tons in 2030, while keep leveling up to 400 million metric tons in 2050[24]. Based on this background, the carbon-neutral production of light olefin and LPG has a great impact and a bright future.

CO$_2$ hydrogenation, which is a crucial catalytic CO$_2$ conversion reaction, can occur through the methanol intermediate or Fischer-Tropsch synthesis (FTS) routes. To our knowledge, almost all the LPG synthesis methods by C1 chemistry until now, regardless of whether they used syngas (CO/H$_2$) or CO$_2$/H$_2$, employed a methanol-intermediated route by combining methanol synthesis catalysts with zeolites[17–20]. Almost no Fischer-Tropsch route was reported for LPG

[1]Department of Applied Chemistry, School of Engineering, University of Toyama, Gofuku 3190, Toyama 930-8555, Japan. [2]School of Materials Science and Engineering, Shenyang University of Chemical Technology, Shenyang, Liaoning 110142, China. [3]School of Chemistry and Chemical Engineering, Anhui University, Hefei, Anhui 230601, China. [4]These authors contributed equally: Jiaming Liang, Jiangtao Liu. ✉e-mail: lsguo@ahu.edu.cn; yasu@eng.u-toyama.ac.jp; liangbing@syuct.edu.cn; tsubaki@eng.u-toyama.ac.jp

synthesis, especially from $CO_2/H_2$. Although methanol as an intermediate pathway can break the Anderson-Schulz-Flory (ASF) distribution and obtain a target product with high selectivity, it often suffers from low $CO_2$ conversion (10–35%) and high CO selectivity (20–75%) due to the thermodynamic equilibrium limitation and thus does not meet the needs of industrial production[2]. On the other hand, $CO_2$ hydrogenation to light olefins is still a hot area, but it faces challenges in increasing $CO_2$ conversion, suppressing CO by-product selectivity, and enhancing light olefin selectivity. Therefore, a modified Fischer-Tropsch route for light olefin and LPG synthesis that simultaneously maintains a high reaction rate and breaks the ASF distribution is urgently needed.

Iron-based catalysts are the most commonly used catalysts for the FTS due to their high reaction activities in both reverse water gas shift (RWGS) and chain growth reactions[25–27]. However, an unmodified iron-based catalyst typically exhibits poor activity and high by-product (CO, $CH_4$, $C_2H_6$, etc.) selectivity[28]. To overcome this issue, alkali metal ions, such as K and Na, were added to boost the $CO_2$ adsorption and the contents of active phases[29,30]. Indeed, these modified iron-based catalysts without the use of zeolites presented comparable catalytic performances to those of the zeolite-containing composite catalysts[25,31].

Besides, works on bimetallic catalysts that combined Fe with other active metal components (Co, Cu, Ni, etc.) have also been investigated. Among them, the incorporation of Co to Fe-based catalysts has been proven to enhance the reactivity and target product selectivity[26,28,29,32,33]. Deo et al. discovered that the addition of controlled amounts of Co to Fe resulted in high yields of methane[34]. Furthermore, Xu et al. proposed that the generation of active iron-cobalt carbides originating from a ternary $ZnCo_xFe_{2-x}O_4$ catalyst was conducive to the formation of light olefins[35]. Recently, our group reported a spinal-like $ZnFe_2O_4$ with a small amount of cobalt incorporation for $CO_2$ conversion and found that the presence of $Co_3Fe_7$ sites could facilitate a high-yield production of liquid fuels (26.7% for $C_5^+$)[36]. Similarly, Zhang et al. detected that the Na-promoted CoFe alloy benefited the formation of jet fuel[37]. These reports manifest that the combination of Fe and Co can be used as a powerful and efficient catalyst for the selective conversion of $CO_2$, and the intimate interaction between cobalt and iron species is able to tune the product distribution. To our knowledge, the supported iron-based catalysts could merely generate one type of hydrocarbon during direct $CO_2$ hydrogenation. However, given the different intrinsic properties of Fe and Co in the formation of hydrocarbon products, where iron contributes to the alkene production and cobalt contributes to the saturated alkane production[38,39], the rational regulation of Fe and Co active site distribution may play a role in transforming the product types. Furthermore, the introduction of support has been revealed to significantly influence the local environment of the active sites. Even a three-dimensional encapsulation structure of graphene led to a fascinating result[40,41]. Based on the above assumptions, the particles of Fe and Co with rational spatial distributions regulated by the graphene support may achieve integrated production of different types of hydrocarbons.

Herein, we report a graphene-fence engineering approach to regulating multiple active sites of Fe-Co bimetallic catalysts for product-switchable $CO_2$ hydrogenation. Taking advantage of the structural transformation of graphene during the reduction process, a series of graphene-supported Fe-Co bimetallic catalysts with different internal and surface distributions of active sites were successfully synthesized. The Fe-Co active sites tuned the demand for carbon chain growth and olefin secondary hydrogenation, leading to an integrated and switchable process for selective $CO_2$ hydrogenation to light olefins or LPG. Iron carbides combined with metallic cobalt on the surface of graphene fences could catalyze $CO_2$ hydrogenation to light olefins (50.1% for $C_2^=-C_4^=$) at a conversion of 55.4%. Whereas the scattered spatial active sites of iron carbides and metallic cobalt, separated by graphene fences, achieved LPG ($C_3^P-C_4^P$) selectivity of 43.6% at a conversion of 46%. Meanwhile, it created a precedent for $CO_2$ hydrogenation to LPG via a Fischer-Tropsch pathway and exhibited an ultra-high STY (space-time yield) of LPG (151.0 g $kg_{cat}^{-1}$ $h^{-1}$), which was much higher than any other previously reported composite methanol-intermediate catalysts (Supplementary Fig. 1). In addition, the graphene fences could also protect the metal particles from being deactivated by agglomeration, thus maintaining high activity for a long time in a continuous test. Our research offers methodologies for manipulating the graphene material as fences to divide active nanoparticles and switch product types and sheds light on the rational design of multiple active sites for the synthesis of target chemicals (Supplementary Fig. 2).

## Results

### Adjustable spatial distribution of multiple sites

A series of graphene-supported Fe-Co bimetallic catalysts with identical total contents of Fe, Co, and K were synthesized by varying the addition order of Fe and Co during the hydrothermal and impregnation processes (Supplementary Fig. 3). Upon examining the as-prepared catalysts, only the characteristic diffraction peaks ascribed to rGO and $Fe_2O_3$ were observed in XRD (Fig. 1a). There were no peaks associated with Co detected in the four graphene-supported catalysts. To estimate the total contents of different metal elements, we carried out the inductively coupled plasma-optical emission spectrometer (ICP-OES) tests and found that the contents of Fe, Co, and K were close to the theoretical values of 20 wt%, 4 wt%, and 1 wt%, respectively (Supplementary Table 1). Although the total contents of each metal were roughly the same for different catalysts, the unique structure separated by graphene fences formed in the hydrothermal process led to different spatial distributions of Fe and Co sites in the inner and surface layers.

Throughout the hydrothermal process, the decrease in graphene layer distances and the cross-linking of the graphene layers led to the dynamic transformation of GO from the 2D lamellar structure to the 3D stereoscopic structure[40,41]. To demonstrate the dynamic evolution, we employed in situ XRD to detect the diffraction peak shift of GO during the temperature-programmed reduction process. With the temperature increasing and $H_2$ introduction, diffraction peaks gradually shifted to the higher direction (Supplementary Fig. 4), representing the decrease in graphene layer spacings as determined by Bragg's law[42]. Similarly, the rGO obtained by hydrothermal treatment also exhibited a higher peak and a smaller layer spacing compared with GO (Fig. 1b). Meanwhile, the specific surface area significantly decreased (Supplementary Table 2). These phenomena corresponded to the folding and bending of the graphene layers, as observed in SEM images (Fig. 1c).

Accordingly, due to the cross-linking effect of the graphene layers during the hydrothermal process, the metals added together with GO were partially encapsulated in the folded inner layers and in situ replaced the oxygen-containing groups of the graphene layers. Consequently, the "graphene fences" were formed by the reduced graphene layers, which encased metal nanoparticles. After the hydrothermal treatment, the metals introduced by impregnation were more easily loaded onto the surface of the folding graphene layers instead of the internal layers, owing to the separation effects of the graphene fences. These unique structures were reflected in the molecular vibration spectra, surface and internal element contents, and morphological characterizations of the catalysts.

We employed the FTIR spectra to identify the loss of oxygen-containing groups replaced by metal sites (Fig. 1d). The stretching vibrations of O – H (3400 $cm^{-1}$), C = O (1722 $cm^{-1}$), aromatic C = C (1620 $cm^{-1}$), carboxyl O = C – O (1356 $cm^{-1}$), epoxyl C – O (1220 $cm^{-1}$), and alkoxyl C – O (1050 $cm^{-1}$), which were recorded as references, were

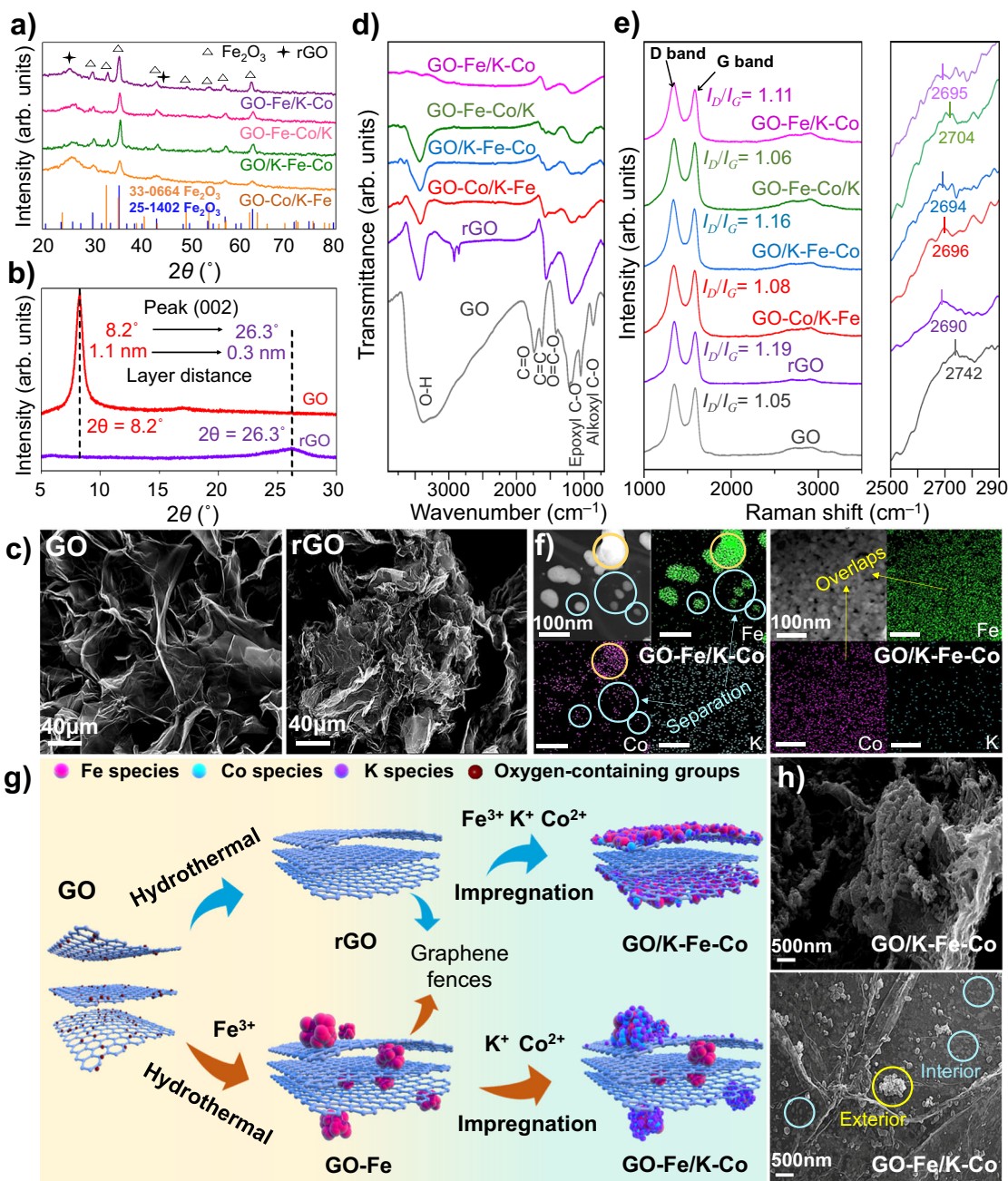

**Fig. 1 | The structure and morphology investigation of graphene-supported Fe-Co catalysts. a** XRD patterns of fresh catalysts. **b** XRD patterns of GO and rGO. **c** SEM images of GO and rGO; the bars stand for 40 μm. **d** FTIR spectra of fresh catalysts. **e** Raman spectra of fresh catalysts. **f** TEM mapping images of GO-Fe/K-Co and GO/K-Fe-Co catalysts; the bars in the images stand for 100 nm. The yellow circles represent the overlapping distribution areas, while the light blue circles represent the separation areas. **g** Schematic diagram of the preparation processes, spatial structures, and species distributions of GO/K-Fe-Co and GO-Fe/K-Co. Fe species, red balls; Co species, light blue balls; K species, purple balls. **h** SEM images of GO/K-Fe-Co and GO-Fe/K-Co catalysts. The light blue and yellow circles represent the metal sites separated by the graphene fences.

all observed in the FTIR spectrum of GO. After hydrothermal synthesis, the oxygen-containing group peaks of rGO were significantly diminished, and two broad peaks at 1550 cm$^{-1}$ and 1175 cm$^{-1}$ appeared, which were assigned to C=C and C−O, respectively[43]. Intriguingly, the vibration peaks of metal-supported catalysts displayed smaller areas compared to those of rGO, especially in GO-Fe/K-Co and GO-Fe-Co/K, which could be interpreted as more substitutions of oxygen-containing groups by metals in the reduction process, caused by the metals loading the graphene inner layers. However, the metals in GO/K-Fe-Co and GO-Co/K-Fe were unable to adequately replace the oxygen-containing groups inside the layers due to the protection of graphene fences, thus resulting in a larger peak area.

Furthermore, the thickness of the graphene-supported Fe-Co bimetallic catalysts could also reveal the various structures. The thicknesses of the graphene were detected by the second-order peak positions of the Raman spectra, which appear near 2700 cm$^{-1}$. In general, a higher peak position represents a bigger graphene thickness and more graphene layers[44,45]. GO and rGO displayed the maximum and minimum graphene thicknesses of graphene layers, respectively (Fig. 1e). With the incorporation of Fe and Co, the thicknesses of graphene increased compared with rGO, among which the GO-Fe-Co/K showed the largest thickness, revealing that the presence of both Fe and Co on the inner layers hindered the compression of the graphene layers during the hydrothermal process. As inferred, GO/K-Fe-Co

showed the smallest thickness of all the graphene-supported catalysts. $I_D/I_G$ values displayed the reverse order as the thicknesses, indicating that the disorders increased when the graphene thicknesses were compressed (Fig. 1e).

The varying Fe-Co distributions were further demonstrated by the different metal contents between the surface and interior. Based on the XPS results, the Fe surface contents in GO-Co/K-Fe (12.3%) and GO-K-Fe-Co (11.7%) catalysts whose Fe was introduced by impregnation were distinctly higher than those of the catalysts whose Fe was loaded by hydrothermal incorporation, such as GO-Fe-Co/K (3.4%) and GO-Fe/K-Co (4.1%) (Supplementary Table 3). Meanwhile, the GO/K-Fe with a higher surface Fe content (Supplementary Table 3) exhibited a stronger magnetism intensity in the M-H loop compared to the GO-Fe/K (Supplementary Fig. 5), which ulteriorly supported the conclusion that Fe added by the hydrothermal process was partially encased in the graphene layers, resulting in a lower surface content and a weaker magnetism intensity. Furthermore, the Fe/Co values depicted by SEM mapping, which also reflected the surface element contents, exhibited the same trend as those measured by XPS (Supplementary Fig. 6 and Supplementary Table 4). In addition to the surface content characterizations, we applied FIB-SEM to directly investigate the metal distributions in the graphene inner layers. In the cross-sectional SEM images of GO-Fe/K-Co, the metal nanoparticles were found to be loaded between the graphene layers (Supplementary Fig. 7a), which corresponded to the oxide and iron mapping distributions (Supplementary Fig. 7b, c), while only a little cobalt distribution was observed on the internal metal particles (Supplementary Fig. 7d). Meanwhile, the Fe/Co value of the cross-section (8.42) obtained by the SEM mapping (Supplementary Table 5) was significantly higher than that of the catalyst surface (0.82) (Supplementary Table 4). The various metal distributions on the surface and interior effectively corroborated the reconstruction of the graphene-supported catalysts, forming a unique structure with different spatial distributions of multiple active sites.

Additionally, as shown in Fig. 1f, the yellow circles displayed the same Fe and Co distributions in GO-Fe/K-Co, while in the red circles only the Fe distribution was clearly observed. This could be explained by the difficulty of impregnating Co in the folded interior graphene layers as compared to Fe loaded by hydrothermal synthesis, thus forming the active sites with different metal compositions. As a result, the yellow and red circles represented the surface and internal metal sites of the graphene layers, respectively. By contrast, Fe and Co in GO/K-Fe-Co exhibited a uniform and well-dispersed distribution over the outer surface of graphene layers (Fig. 1f). These findings convincingly verified that the spatial distributions of Fe-Co active sites could be regulated by the graphene fences, forming scattered sites (Fe+CoFe) or uniform assemblage sites (FeCo) (Fig. 1f).

In response to the descriptions mentioned above, a detailed schematic diagram of dynamic evolution regarding the synthesis of GO/K-Fe-Co and GO-Fe/K-Co was drawn in Fig. 1g. In terms of the GO/K-Fe-Co catalyst, GO was first reduced by the hydrothermal process, leading the graphene layers to be twisted and folded, thus forming graphene fences. Subsequently, Fe and Co were incorporated through impregnation and primarily dispersed on the surface of the graphene fences with a benign distribution, on account of the resistance of the folded interior layers. In the case of the GO-Fe/K-Co catalyst, Fe was introduced during the hydrothermal process, which would in situ replace the oxygen-containing groups and be partially encapsulated in the interior graphene layers. Then the impregnated Co was loaded on the exterior graphene layers because of the inaccessible accesses controlled by the graphene fences, forming scattered sites with a small amount of Fe on the outer surface of the graphene layers. The two different spatial distribution structures described above could be directly observed by the SEM images. Figure 1h clearly showed Fe-Co homogeneity active sites arranged uniformly on the surface of graphene layers (GO/K-Fe-Co) and the metal sites separated by graphene fences (GO-Fe/K-Co), in which the yellow circle represented the sites on the surface layer and the red circles represented the sites on the internal layer. Therefore, with the assistance of developed graphene fences, graphene-supported Fe-Co bimetallic catalysts with adjustable metal spatial distributions in the internal and external layers were successfully synthesized.

## Phase composition characterizations of Fe-Co catalysts

Several characterizations were first applied to determine the surface metal phase compositions of the as-prepared catalysts. The HR-TEM images (Supplementary Fig. 8) revealed the lattice spacings of 0.25 nm that corresponded to the (119) plane of the $Fe_2O_3$ species in the four graphene-supported catalysts, which was consistent with the XRD results shown in Fig. 1a[46]. The XPS Fe 2p spectra (Supplementary Fig. 9) showed binding energy peaks at 710.8 eV and 712.8 eV, which were ascribed to the Fe(II) and Fe(III) phases, respectively[47]. Moreover, according to the $^{57}$Fe Mössbauer spectra (Supplementary Fig. 10 and Supplementary Table 6), Fe was predominantly present in the form of $Fe_3O_4$, which was the combination of $Fe^{2+}$ and $Fe^{3+}$ species. Due to the low contents or benign distributions, there were no diffraction peaks attributed to Co in XRD patterns, whereas two peaks assigned to $Co^{2+}/Co^{3+}$ and $Co^0$ appeared in the Co 2p XPS spectra (Supplementary Fig. 11)[48]. To further ascertain the Co proportion on the catalytic surface, XANES spectra with fitting curves were recorded. As pictured in Supplementary Fig. 12, CoO was identified as the predominant phase of Co species regardless of their spatial distributions. Besides, the Co K-edge EXAFS curve fitting results and parameters were also displayed in Supplementary Fig. 13 and Supplementary Table 7, disclosing the existence of the Co–Co and Co–O bonds.

Regarding the spent catalysts, the Fe 2p spectra showed binding energy peaks at 708.5 eV, which were assigned to the Fe-C bonds (Fig. 2b), indicating the presence of iron carbides[49,50]. Based on the results of XRD patterns, diffraction peaks attributed to $Fe_5C_2$ were observed in all spent catalysts (Fig. 2a). Among them, the diffraction peaks with the strongest intensities corresponded to $Fe_5C_2$ (510) facets, which were also revealed by HR-TEM images with lattice spacings of 0.20 nm (Supplementary Fig. 14)[51]. Furthermore, the coexistence of $Fe_5C_2$ (A) and $Fe_5C_2$ (B) species was determined from two overlapping sextuplets in the $^{57}$Fe Mössbauer spectra (Fig. 2c and Supplementary Table 8), which represented the different occupied sites of Fe in the crystallographic structure of $Fe_5C_2$[52]. The aforementioned findings demonstrated that iron carbide, an active phase for the chain growth reaction, was mainly presented as the $Fe_5C_2$ (510) phase in the spent catalysts[21]. Co K-edge XANES tests were also applied to identify the Co phases in the spent catalysts. In contrast to the as-prepared catalysts, the majority of cobalt existed in metallic states (Fig. 2d), demonstrating the good reduction abilities of Co in the graphene-supported catalysts. Moreover, the EXAFS fitting results illustrated the presence of only Co–Co bonds, providing further evidence that cobalt existed in the metallic phases (Supplementary Fig. 15 and Supplementary Table 9).

## Catalytic performances and stabilities

The catalysts were tested under the conditions of 320 °C, 3.0 MPa, and W/F = 4.5 g h mol$^{-1}$. The main products of both the GO-Fe/K and GO/K-Fe catalysts, without the addition of Co, were methane and light olefins instead of saturated light paraffins (Fig. 3a). However, as shown in Fig. 3b, with the Co incorporation, the Fe-Co bimetallic catalysts displayed various types of hydrocarbon product selectivities. GO-Co/K-Fe and GO/K-Fe-Co mainly produced light olefins, especially for the GO/K-Fe-Co catalyst, where 50.1% $C_2^=$–$C_4^=$ selectivity was obtained at a $CO_2$ conversion of 55.4%. Whereas for GO-Fe-Co/K, more alkanes, particularly methane and ethane, were produced (Fig. 3b). However,

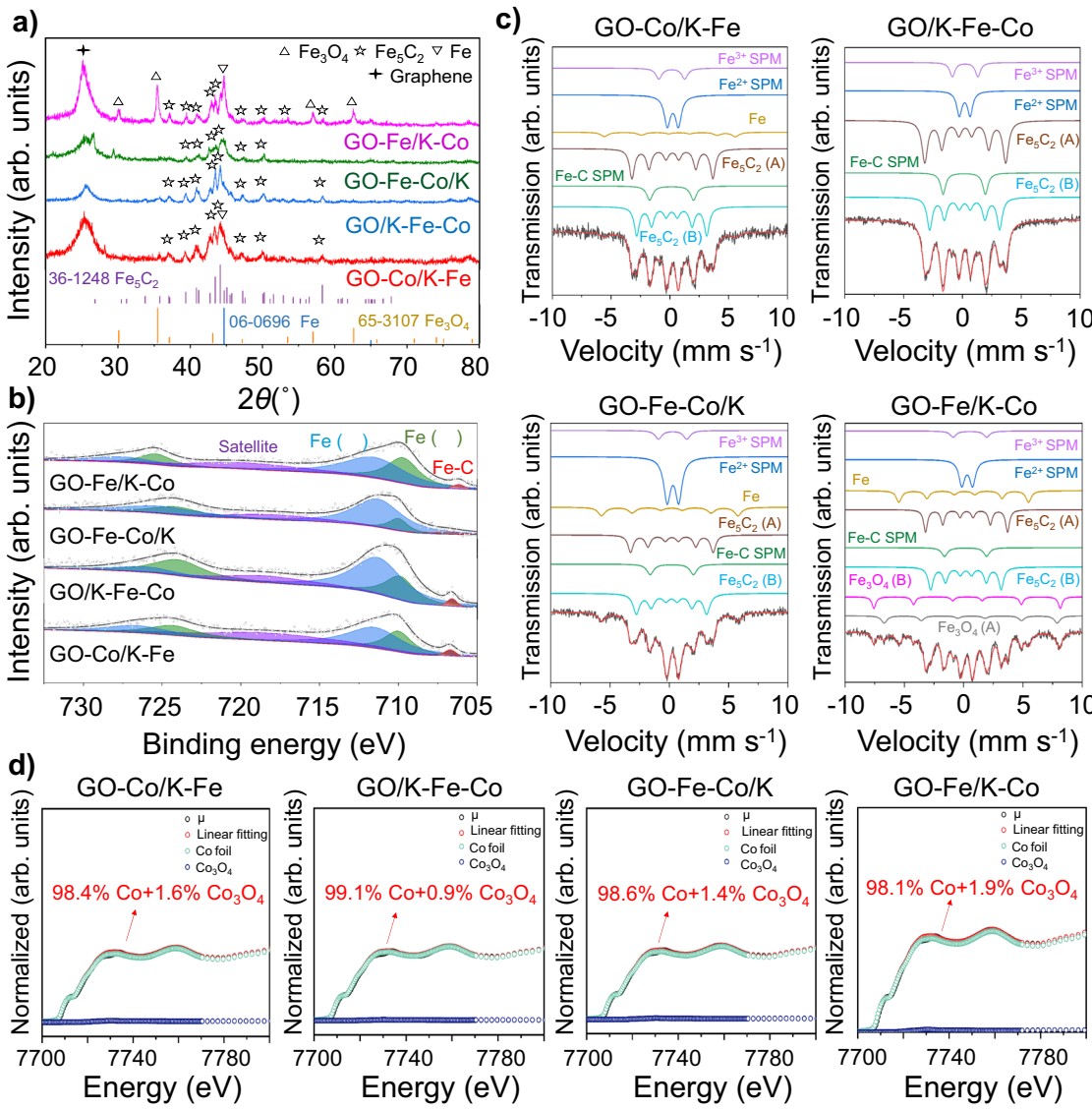

**Fig. 2 | Phase compositions of the spent graphene-supported Fe-Co bimetallic catalysts. a** XRD patterns. **b** Fe 2*p* XPS spectra. **c** $^{57}$Fe Mössbauer spectra. **d** Co K-edge XANES spectra with fitting curves.

GO-Fe/K-Co exhibited a higher selectivity of propane and butane (43.6%) compared to the GO-Fe-Co/K (Fig. 3b). In comparison with GO-Fe/K, the CoFe sites of GO-Fe/K-Co were formed by impregnating Co onto the surface Fe sites of GO-Fe/K with the assistance of graphene fences. GO/K-10Fe-20Co was also employed to simulate the external CoFe sites of GO-Fe/K-Co which had similar surface Fe/Co ratios (0.6 and 0.7) (Supplementary Table 3). The high methane selectivity confirmed that the external CoFe sites with a high Co content had a strong hydrogenation capacity (Fig. 3a), which facilitated the original light olefins produced on the internal Fe sites hydrogenating into saturated alkanes, thus transforming the products from light olefins to LPG (Fig. 3g). In this process, the reaction equilibrium was shifted in the positive direction, leading to an increase in the $CO_2$ conversion from 33.2% (GO-Fe/K) to 46.0% (GO-Fe/K-Co). This can also be observed in the detailed product distributions of GO-Fe/K and GO-Fe/K-Co (Supplementary Fig. 16). Before the Co addition, the product selectivity of C2 and C3 in GO-Fe/K was roughly the same. However, after the introduction of Co, C3 products occupied the highest selectivity, which further indicated that the diffusion and hydrogenation effects on the internal-active-site products shifted the chemical equilibrium in a positive direction.

In contrast to the GO-Fe/K-Co, the products of the rGO-Fe/K-Co were mainly light olefins rather than alkanes (Fig. 3a). This was because the absence of the spatial dual active sites separated by the graphene fences (Supplementary Fig. 17), which is proven by Supplementary Fig. 18 (same distributions of Fe and Co in TEM elemental mapping images), resulted in a higher Fe/Co ratio (Supplementary Table 3), thus weakening the hydrogenation ability. Such a result further proved that the unique spatial distributions of Fe-Co dual active sites tuned by graphene fences could efficiently control product types.

By comparing the TEM images (Fig. 1f and Supplementary Figs. 19–21), we observed that the fresh GO-Fe-Co/K and GO/K-Fe-Co catalysts presented smaller particle sizes compared to the other catalysts, indicating that the simultaneous addition of Fe and Co constrained the aggregation of Fe nanoparticles and enhanced the dispersions of Fe species. $CO_2$ hydrogenation is a structurally sensitive reaction, and higher dispersions are beneficial for improving the carbonization ability of Fe and thus enhancing the catalytic performance (Fig. 3g)[53]. As seen, according to the $^{57}$Fe Mössbauer spectra of the spent catalysts (Fig. 2c and Supplementary Table 8), GO/K-Fe-Co exhibited the largest proportion of $Fe_5C_2$ (72%), including both $Fe_5C_2$ (A) and $Fe_5C_2$ (B). Accordingly, GO/K-Fe-Co also manifested the highest

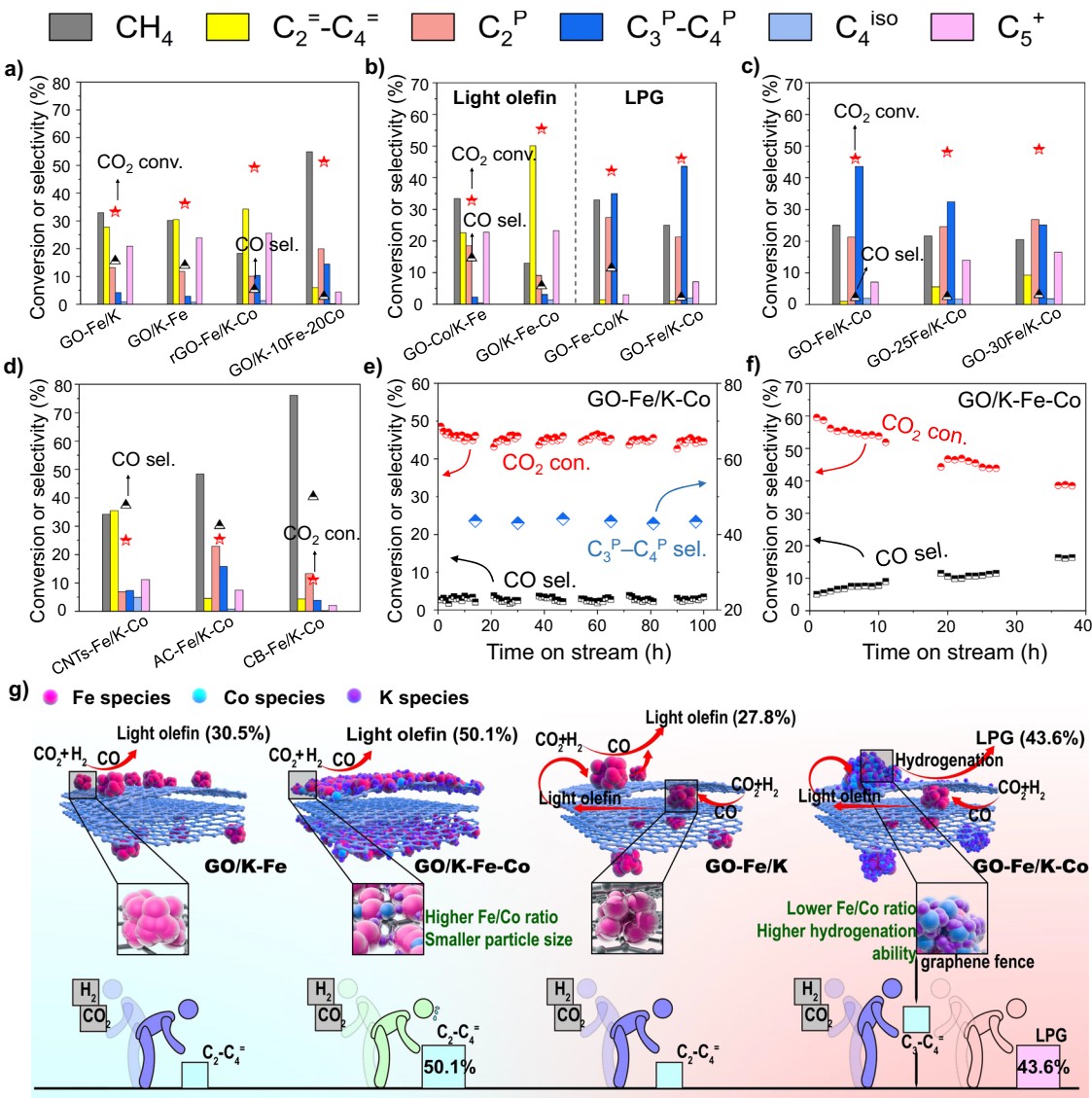

**Fig. 3 | Catalytic performances and stabilities. a–d** Catalytic performances of the catalysts; Reaction conditions: 320 °C, 3.0 MPa, TOS = 8 h, W/F = 4.5 g h mol⁻¹, $H_2$/$CO_2$/Ar = 68: 27: 5. **e–f** Stabilities of GO-Fe/K-Co (**e**) and GO/K-Fe-Co (**f**). **g** Schematic diagrams of the graphene fences regulated Fe-Co active sites controlling product type. Fe species, red balls; Co species, light blue balls; K species, purple balls. Characters in varied colors represent varied active sites in the frame.

$CO_2$ conversion (55.4%) and rather high light olefin selectivity (50.1%) (Fig. 3b). The GO-Fe-Co/K catalyst, however, displayed the highest temperature of peaks in the $H_2$-TPR (peak γ) and $CO_2$-TPD profiles (Supplementary Figs. 22 and 23). This phenomenon could be explained by the fact that smaller particles would enter or penetrate the interlayers more easily during the hydrothermal process, which was proved by the smaller surface metal loadings, as shown in Supplementary Table 3. As a result, the metal nanoparticles tightly covered by graphene layers showed strong metal-support interaction (SMSI), which made the reduction and carbonization more challenging[53]. This view was intuitively illustrated by in situ XRD. When the temperature rose to 400 °C, a peak of metallic iron appeared on the GO-Fe-Co/K catalyst. Obviously, this temperature was the highest of all the catalysts. Correspondingly, in the carbonization process, GO-Fe-Co/K also showed the lowest peak strength of $Fe_5C_2$ (Supplementary Fig. 24). Meanwhile, previous research has demonstrated that strong chemical adsorption of $CO_2$ is not easily triggered for hydrogenation, and the coating of the surface Fe species formed by the strong chemical adsorption of $CO_2$ may potentially lower the catalytic performance[38]. Consequently, these reasons led to a slightly lower $Fe_5C_2$ content (44.6%) and $CO_2$

conversion of GO-Fe-Co/K (42.1%) than those of GO/K-Fe-Co (72% and 55.4%) (Fig. 3b and Supplementary Table 8).

In contrast to GO-Fe-Co/K, the GO-Fe/K-Co catalyst displayed appropriate metal-support interaction (MSI) and $CO_2$ adsorption strengths (Supplementary Fig. 22 and Supplementary Fig. 23), leading to a $CO_2$ conversion of up to 46% (Fig. 3b). Given that cobalt was inactive in the RWGS reaction[39,54], the Co surrounding the $Fe_5C_2$ on the surface layers would further consume CO without producing it[55]. Accordingly, GO/K-10Fe-20Co, which was used to simulate the external Fe/Co sites, also exhibited a low CO selectivity (2.7%). It was significantly lowered compared to that of the GO-Fe/K (15.6%), which consisted of Fe active sites (Fig. 3a). This finding further suggested that the CoFe active sites with a low Fe/Co ratio (0.82) (Supplementary Table 4) located on the external surface of GO-Fe/K-Co could consume the CO produced by the internal Fe sites, thus keeping the CO selectivity at a low level. As a result, GO-Fe/K-Co showed an ultra-low CO selectivity (2.2%) among all the catalysts (Fig. 3b). To the best of our knowledge, this is the lowest value level reported for the current methanol intermediate route as well as a modified FTS pathway (refer to Fe-containing catalysts). Furthermore, thick carbon deposition

around metal particles would reduce the catalytic activity[56,57] and this phenomenon can be observed in TEM images of the spent catalysts (Supplementary Fig. 14). Clearly, as determined by TEM mapping, the ratio of Fe/Co in the spent GO-Co/K-Fe was much lower than that of other catalysts and the fresh GO-Co/K-Fe (Supplementary Fig. 25, Supplementary Table 10 and Supplementary Table 11), which can be explained by the large amount of amorphous carbon deposited on the surface of Fe sites affecting the determination of element contents. As a result, it exhibited a relatively low $CO_2$ conversion (32.8%) (Fig. 3b).

To further investigate the influence of the Fe amount, the total Fe content in the GO-nFe/K-Co catalysts was altered from 20 wt% to 30 wt%. Intuitively, as Fe content rose, the selectivity of LPG ($C_3^P$–$C_4^P$) declined, while the selectivities of light olefins ($C_2^=$–$C_4^=$) and $C_5^+$ products increased (Fig. 3c), revealing that the extra Fe introduced to the interior and external graphene fences resulted in an enhancement of the carbon chain growth capacity and an inhibition of the olefin secondary hydrogenation, respectively. In addition, to compare the performances of GO-Fe/K-Co with other carbon materials (CNTs, AC, and CB) supported catalysts, the same preparation procedures and addition amounts of GO-Fe/K-Co were performed. The LPG selectivities of AC-Fe/K-Co (15.8%) and CB-Fe/K-Co (3.9%) were significantly lower than those of GO-Fe/K-Co (43.6%) due to the excessive formation of by-product methane and ethane. Besides, the main products of CNTs-Fe/K-Co were light olefins (35.5%) rather than $C_3$-$C_4$ saturated alkanes (7.3%) (Fig. 3d). Meanwhile, the metal distributions of these catalysts were determined by TEM elemental mapping images (Supplementary Fig. 26). Co distributions were found in all the Fe distribution areas in these three catalysts, indicating that these three carbon materials lacked the function of separating metals as graphene. These findings further demonstrated the superior performance of graphene-fence-separated dual active sites in regulating carbon chain growth and olefin secondary hydrogenation.

We further investigated the stabilities of GO-Fe/K-Co and GO/K-Fe-Co under 3.0 MPa with a W/F of 4.5 g h mol$^{-1}$ at 320 °C. For GO/K-Fe-Co, the $CO_2$ conversion decreased and the selectivity of CO as a byproduct increased continuously within 40 h on stream. By contrast, GO-Fe/K-Co remained stable during the 100-hour stability test (Fig. 3e, f). Interestingly, as depicted in TEM images (Supplementary Fig. 19a), the metal nanoparticles of the spent GO/K-Fe-Co catalyst aggregated dramatically in comparison to the fresh catalyst with the prolongation of the reaction process. Contrarily, the metal particles of the spent GO-Fe/K-Co did not agglomerate, and the particle sizes maintained a stable range throughout the reaction (Supplementary Fig. 19a) due to the protection of the graphene fences. Previous studies have demonstrated that the graphene loaded with iron nanoparticles by the hydrothermal method has a role in anchoring iron particles[41]. Combined with our results described above, the graphene fences that separated the dual active sites in the GO-Fe/K-Co catalyst could also spatially confine the aggregation of metal particles during the reaction, thus preventing the deactivation and maintaining a high activity (Supplementary Fig. 19b). Besides, TEM mapping images of the spent GO-Fe/K-Co catalysts were also tested to explore the distributions of Fe and Co after the reaction, and enlarged images are shown in Supplementary Fig. 27. The red circles represent the area with different distributions of Fe and Co (obvious Fe distributions but few Co distributions). As observed, after the reaction process, graphene fences still maintain the effect of separating Fe and Co active sites.

## Mechanistic studies

According to the local micro-environments of supported metal catalysts reported in the previous work[31], in this study, we employed density functional theory (DFT) calculations to explore the influence of the Fe-Co dual sites separated by graphene fences. To precisely identify the micro-environment of each active site, we first constructed a

$Fe_5C_2$ (510) surface model, which was denoted as Model 1. Afterwards, Model 2 was built by adding a $Co_{10}$ cluster to Model 1 in consideration of surface cobalt incorporation of GO-Fe/K-Co (Fig. 4b). Thereinto, Model 1 corresponded to the $Fe_5C_2$ active sites inside the graphene fences of GO-Fe/K-Co, whereas Model 2 represented the $Fe_5C_2$/Co active sites on the surface of graphene fences (Fig. 4b). In order to compare the adsorption capacities of Co carbide and metallic Co, $Co_{10}$ in Model 2 was substituted by $Co_8C_4$, and it was dubbed "Model 3" (Fig. 4b). According to the adsorption energy results, the adsorption of both the $H_2$ and light olefins on the surface cobalt sites in Model 2 and Model 3 was more stable than that on the interfacial sites, illustrating that $H_2$ and light olefins were primarily adsorbed on the surface cobalt sites of $Fe_5C_2$/Co due to the electron transfer between cobalt and iron carbide (Fig. 4c, d, and Supplementary Fig. 28). The electron transfer between iron carbide and cobalt on Model 2 could be intuitively observed by the charge density difference analysis, which was shown in Supplementary Fig. 29. The red and green colors represented electron accumulation and loss, respectively. Obviously, after being loaded onto $Fe_5C_2$, electrons were transferred from cobalt to $Fe_5C_2$. Since electron-deficient metals are more favorable to adsorb hydrogen[58], it was the electron transfer that enhanced hydrogen adsorption on the cobalt site, while hydrogen was not likely to be adsorbed on the interface of $Fe_5C_2$-Co due to the electron accumulation.

Regarding the hydrogen adsorption, the hydrogen adsorption over Model 2, which adsorbed at the surface metallic Co sites, had a lower adsorption energy (−0.84 eV) than that over iron carbide in Model 1 (−0.81 eV), revealing that the $Fe_5C_2$/Co sites on the external graphene fences had a stronger $H_2$ adsorption effect than that of the $Fe_5C_2$ sites inside the graphene fences (Fig. 4c). Accordingly, the $H_2$ temperature-programmed desorption profile ($H_2$-TPD) of GO-Fe/K-Co exhibited two distinct peaks (I and II) (Fig. 4e), which corresponded to the weak chemical adsorption and strong chemical adsorption of hydrogen, respectively. rGO-Fe/K-Co displayed a desorption peak near 600 °C resembling that of GO-Fe/K-Co. However, unlike GO-Fe/K-Co, the strong chemical adsorption peak around 650 °C was not obvious (Supplementary Fig. 30), and this was because partial reduction of rGO-Fe/K-Co had already been accomplished when Fe was introduced, making graphene fences ineffective at dividing the dual active sites. Moreover, the GO-Fe/K catalyst also displayed a single weak hydrogen chemisorption peak before Co was added (Supplementary Fig. 30). This evidence strongly proved that the different hydrogen adsorption capacities between the internal $Fe_5C_2$ sites and the surface $Fe_5C_2$/Co sites after the Co addition were the main reasons for the formation of the various hydrogen desorption peaks of GO-Fe/K-Co in $H_2$-TPD. Wherein $Fe_5C_2$ corresponded to the weak hydrogen chemisorption peak (peak I), while $Fe_5C_2$/Co corresponded to the strong hydrogen chemisorption peak (peak II) (Fig. 4b, c, and e).

Meanwhile, GO-Co/K-Fe presented one peak in the $H_2$-TPD profile as well, which could be explained by the fact that a large quantity of Fe was loaded on the surface of the metallic Co, thus inhibiting the hydrogen adsorption on the Co sites. Furthermore, Fe and Co were distributed evenly and compactly on the graphene fences in GO-Fe-Co/K and GO/K-Fe-Co due to their simultaneous addition, hence they also presented a single chemisorption peak (Fig. 4e). Among them, GO-Fe-Co/K exhibited the highest Co valence states in the XANES results (Supplementary Fig. 12), indicating that Co lost the most electrons in GO-Fe-Co/K, while GO-Fe/K-Co, whose Co was supported on the surface Fe, showed the lowest valence states (Supplementary Fig. 12), which can be interpreted as the strong metal-support interaction of GO-Fe-Co/K enhancing the electron transfers between Co and graphene (Supplementary Fig. 22). Past studies have revealed that electron-deficient metals are more likely to absorb hydrogen[58]. Therefore, GO-Fe-Co/K displayed the strongest hydrogen adsorption peak. On the contrary, GO/K-Fe-Co, which was loaded with Fe and Co by impregnation, showed a lower-strength hydrogen adsorption peak

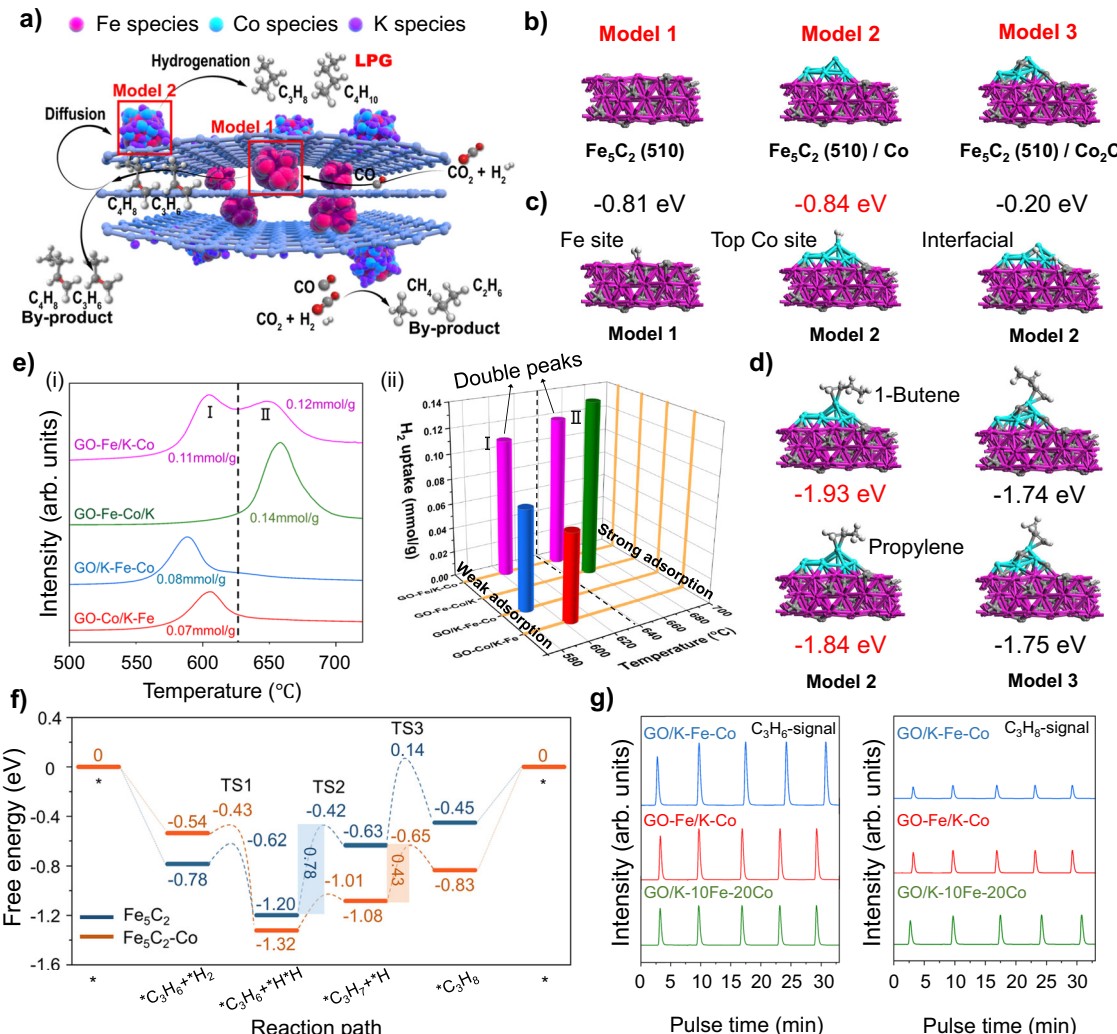

**Fig. 4 | The exploration of the reaction mechanism and path. a** Reaction path diagram. Fe species, red balls; Co species, light blue balls; K species, purple balls. **b** DFT calculation models of $Fe_5C_2$, $Fe_5C_2/Co$, and $Fe_5C_2/Co_2C$. Substrate colors: Fe, pink; C, brown; Co, blue. **c** $H_2$ adsorption models and adsorption energies over $Fe_5C_2$ and $Fe_5C_2/Co$ models. Substrate colors: Fe, pink; C, brown; Co, blue; H, white. **d** Propylene and 1-Butene adsorption models and adsorption energies over $Fe_5C_2$/ Co and $Fe_5C_2/Co_2C$ models. **e** $H_2$-TPD profile (**i**) and three-dimensional $H_2$-TPD diagram (**ii**). **f** DFT calculated proposed catalytic reaction pathway for propylene hydrogenation over $Fe_5C_2$ site and $Fe_5C_2$-Co site in the conditions of 320 °C. The reaction free energies (eV) are shown in the inset. The shaded parts represented the energy barriers of the speed-determining step. **g** $C_3H_6$-pulse transient hydrogenation profiles of the spent catalysts ($C_3H_6$ signal and $C_3H_8$ signal).

(Fig. 4e) due to the weaker metal-support interaction (Supplementary Fig. 22) and fewer electron transfers between metals and graphene (Supplementary Fig. 12). Besides, the $H_2$-TPD test performed on graphene oxide (GO) was applied to exclude the effect of carbon material decomposition. No obvious peaks were observed in the profile, indicating that GO remained stable in the helium atmosphere below 800 °C (Supplementary Fig. 31).

These hydrogen adsorption characteristics (Fig. 4e) were consistent with the catalytic performances (Fig. 3b): weakly chemisorbed hydrogen was easier to be activated and therefore more inclined to hydrogenate $CO_2$ to extend the carbon chains than olefin secondary hydrogenation, thus generating more olefins. Consequently, the primary products of the GO/K-Fe-Co and the GO-Co/K-Fe were light olefins (Figs. 3b and 4e). Among them, GO/K-Fe-Co exhibited a higher $C_2^=$–$C_4^=$ selectivity (50.1%) (Fig. 3b) because of its lower adsorption peak position (Fig. 4e). On the contrary, strongly chemisorbed $H_2$, which was not activated, tended to hydrogenate olefins to manufacture alkanes[58–60], so GO-Fe-Co/K produced more paraffins, especially methane and ethane (Fig. 3b). However, unlike other catalysts, the GO-Fe/K-Co, which presented double peaks in the $H_2$-TPD profile

(Fig. 4e), demonstrated that both the carbon chain growth ability and the olefin secondary hydrogenation ability were attained as mentioned above. As a result, unlike GO-Fe-Co/K, whose main products were methane and ethane, the products in GO-Fe/K-Co were extended and concentrated in propane and butane (43.6%) (Fig. 3b).

In order to clearly verify the differences in the difficulties of the olefin hydrogenation reaction over the spatial dual active sites, on the basis of the structures of Model 1 and Model 2 constructed above, DFT calculations for potential reaction pathways and intermediates of propylene hydrogenation to propane were conducted and summarized (Fig. 4f, Supplementary Figs. 32 and 33)[61]. $Fe_5C_2$ sites needed to overcome an energy barrier of 0.78 eV to convert the *$C_3H_6$ intermediate into *$C_3H_7$. However, for the $Fe_5C_2/Co$ site, the rate-determining step of the whole process was changed to the step of *$C_3H_7$ to *$C_3H_8$. The lower rate-determining energy barrier (0.43 eV) of the $Fe_5C_2/Co$ site indicated a higher propylene hydrogenation activity compared to the $Fe_5C_2$ site (Fig. 4f)[62]. In addition, $C_3H_6$-pulse transient hydrogenation experiments performed on the spent GO-Fe/K-Co, GO/ K-Fe-Co, and GO/K-10Fe-20Co catalysts were applied to realistically examine the propylene secondary hydrogenation capacities on

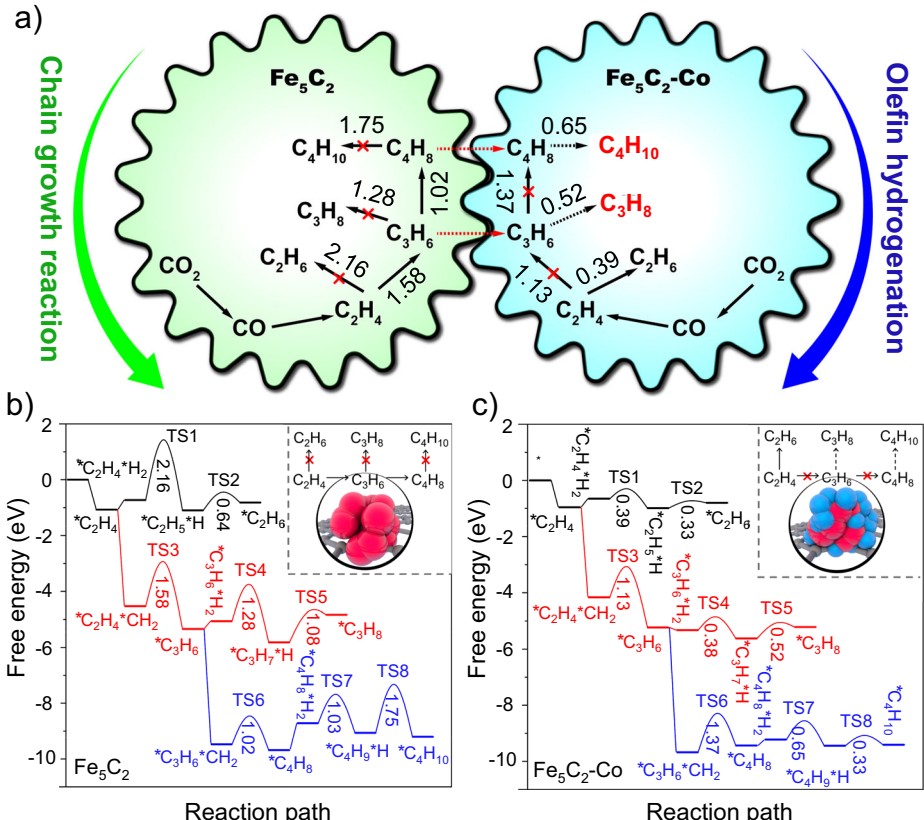

**Fig. 5 | The designed Fischer-Tropsch pathway for CO$_2$ hydrogenation to LPG.**
**a** Synergistic effect of dual active sites for producing LPG from CO$_2$ hydrogenation.
The numbers represent the free energy barriers (eV). **b**–**c** DFT calculations of chain
growth and olefin hydrogenation reactions from ethylene to propane over the
Fe$_5$C$_2$ site (**b**) and the Fe$_5$C$_2$-Co site (**c**). Red and blue balls inside represent Fe and
Co species, respectively.

different active sites. As calculated by DFT, with the same amount of
C$_3$H$_6$ being pulsed into the reactor, GO-Fe/K-Co exhibited a higher
C$_3$H$_8$ signal compared to GO/K-Fe-Co, which demonstrated a stronger
propylene secondary hydrogenation capacity. Furthermore, GO/K-
10Fe-20Co showed the highest C$_3$H$_8$ signal, and this result further
revealed that the external Fe/Co sites were the main active sites for the
propylene hydrogenation reaction to obtain propane (Fig. 4g). We
didn't find pulse peaks attributed to the methane signal in the three
spent catalysts (Supplementary Fig. 34), illustrating that no hydro-
cracking reaction occurred.

Consequently, a reaction path for selective CO$_2$ hydrogenation
over the GO-Fe/K-Co catalyst was proposed in Fig. 4a by analyzing
the reaction results and the mechanism characterizations. Initially,
CO$_2$ was converted into CO via the RWGS reaction at the internal Fe
active sites, followed by the FTS process to produce light olefins. The
diffusion effect could easily transport light olefins to the CoFe active
sites on the external surface of the graphene fences, where the CoFe
sites had a relatively high cobalt content. Due to their higher
hydrogen adsorption capacities, the olefins were hydrogenated to
the light alkanes, producing the high selectivity of LPG. Notably, the
adsorption strengths of the propylene and 1-butene adsorbed on the
metallic Co surface in Model 2 were stronger than those absorbed on
Co$_2$C in Model 3 (Fig. 4d), verifying a higher olefin hydrogenation
efficiency of the metallic Co over Co$_2$C in this unique CO$_2$ hydro-
genation reaction system. Meanwhile, due to the inactivity of Co in
the RWGS reaction, cobalt loaded on the outer surface would con-
sume a large amount of CO produced by the internal Fe active sites,
resulting in an ultra-low CO selectivity (2.2%). However, the high H$_2$
concentration resulting from the lack of RWGS reaction compelled
CO$_2$ hydrogenation directly to the by-products methane and ethane
at the Fe$_5$C$_2$/Co sites on the outer surface of the graphene fences, as

demonstrated by the catalytic performance of GO/K-10Fe-20Co
(Fig. 3a)[39].

In order to clearly explore the synergistic effect of the dual active
sites in the reaction process, we performed DFT calculations for the
chain growth and the olefin hydrogenation reactions from ethylene to
butane over the Fe$_5$C$_2$ and Fe$_5$C$_2$-Co sites (Fig. 5, Supplementary
Figs. 35–37). At the Fe$_5$C$_2$ site, alkenes are more likely to undergo C–C
coupling reactions to achieve carbon chain growth than secondary
hydrogenation reactions due to their lower free energy barriers.
Therefore, more long-chain alkenes would be obtained. For the Fe$_5$C$_2$-
Co site, the hydrogenation of alkenes to alkanes is easier than the chain
growth reaction, so there would be more ethane than propane and
butane produced. However, once the propylene and butene products
diffused from the Fe$_5$C$_2$ sites to the Fe$_5$C$_2$-Co sites, due to the low
energy barriers of the propylene and butene hydrogenation reactions
(0.52 and 0.65 eV), propylene and butene would be easily hydro-
genated to propane and butane, resulting in a high selectivity of LPG
products.

These calculation results also revealed the difficulty of producing
LPG from CO$_2$ hydrogenation via a Fischer-Tropsch pathway, that is,
the contradiction between the carbon chain growth and the olefin
secondary hydrogenation. For the active sites with weak hydrogen
adsorption capacity, such as Fe$_5$C$_2$, it is difficult for alkenes to be
hydrogenated to alkanes, leading to low alkane selectivity. Whereas for
the sites with strong hydrogen adsorption capacity, such as Fe$_5$C$_2$-Co,
on the other hand, it is also difficult to achieve carbon growth, and
excessive methane and ethane products reduce the LPG selectivity
(Figs. 4c and 5). In this situation, the proposed graphene-fence-
separated dual active sites could simultaneously meet the demands of
carbon chain growth and olefin hydrogenation, thus overcoming this
difficulty.

As summarized above, as a catalyst for producing olefins, GO/K-Fe-Co had a surface with a significant number of iron carbide active sites combined with a small number of metallic cobalt sites. The intimate contact (Fig. 1f) and electron transfers between Fe and Co make their hydrogen adsorption capacity tend to be uniform. Meanwhile, due to the lack of separation effects, the higher surface Fe/Co value (Supplementary Table 3) also reduced their hydrogen adsorption capacity[63–65]. Obviously, GO/K-Fe-Co did not exhibit a strong hydrogen chemisorption peak (Fig. 4e) in the condition of such a high Fe/Co ratio on the surface (Supplementary Table 3). Furthermore, as observed in the TEM images (Fig. 1f), K was uniformly distributed on the nanoparticles of Fe and Co due to the well dispersion. Potassium has been proven to restrain $H_2$ chemisorption and increase olefin selectivity[66–68]. Thus, without the assistance of graphene fences, the formed light olefins were easily diffused into the gas flow and carried out due to the weak olefin secondary hydrogenation capacities (Fig. 4e), and exhibited a higher light olefin selectivity (50.1%) (Fig. 3b).

## Discussion

In summary, graphene-fence-regulated Fe-Co bimetallic catalysts with homogeneous active sites or scattered spatial dual active sites were successfully prepared and employed in the $CO_2$ hydrogenation reaction for the selective production of light olefins or LPG without any post-treatments. The GO/K-Fe-Co catalyst, with its uniform distribution and smaller particle sizes, reached a $C_2^=-C_4^=$ selectivity as high as 50.1% at a $CO_2$ conversion of 55.4%. While the graphene-fence-separated GO-Fe/K-Co catalyst displayed a 43.6% selectivity for LPG (propane and butane) and an ultra-low CO selectivity of 2.2% at a 46% $CO_2$ conversion without the help of any zeolites. This reaction result corresponded to the highest STY (151.0 g $kg_{cat}^{-1}$ $h^{-1}$) of LPG ever reported. Characterization and theoretical calculation results demonstrated that the dual active sites (iron carbides and metallic cobalt) performed their tailor-made respective functions, simultaneously satisfying the requirements of suitable carbon chain growth and olefin secondary hydrogenation, and selectively improving the LPG selectivity. Furthermore, the graphene fences prevented the metal particles from agglomerating, thus enhancing the catalytic stability. We expect that this sophisticated method of exploiting the specific structure of graphene to fabricate catalysts with multiple active sites will inspire other important catalytic reactions. Finally, this work not only provides a multiple-active-site catalyst with a unique spatial distribution for selective $CO_2$ hydrogenation but also provides a fundamental understanding of the role of graphene fences in selective hydrogenation. It can be broadened to other supported catalysts and offers valuable guidance for the rational design of powerful reaction environments through engineering the spatial distributions of different active sites.

## Methods

### Catalyst preparation

**Graphene oxide (GO).** $K_2S_2O_8$ (7.5 g, Rgent Chemical Reagent Co.) and $P_2O_5$ (7.5 g, Damao Chemical Reagent Co.) were added to a round-bottomed flask under the conditions of an 80 °C water bath and then combined with concentrated $H_2SO_4$ (Fuchen Chemical Reagent Co.) by stirring for 15 min. Graphite powder (10 g, Sinopharm Chemical Reagent Co.) was subsequently added to the solution. After stirring for 4.5 h, the mixture was filtered and rinsed until the pH of the supernatant reached 7, before being dried overnight. The dried pre-oxidized graphite was then transferred into a three-necked flask with concentrated $H_2SO_4$ in an ice-water bath. Under the aforementioned conditions, $KMnO_4$ (50 g, Sinopharm Chemical Reagent Co.) was added in these preparation phases. The solution was stirred at 35°C for 3 h before being progressively combined with deionized water and 30% $H_2O_2$ until no bubbles occurred, and then aged overnight. The bottom slurry of the solution was transferred to the 3% HCl (Fuchen

Chemical Reagent Co.) for acidification treatment. After filtering and washing to neutrality, GO was then moved to deionized water and ultrasonically agitated for 5 h. Finally, dried graphene oxide was obtained using a freeze-drying method.

**Graphene-supported Fe-Co bimetallic catalysts.** Graphene-supported K-Fe-Co bimetallic catalysts were synthesized using one-pot hydrothermal synthesis and impregnation. The target loadings for the prepared catalysts were 20% Fe, 4% Co, and 1% K, respectively. The ingredients of the obtained catalysts were investigated by inductively coupled plasma optical emission spectrometer (ICP-OES) tests, and the results are shown in Supplementary Table 1.

In detail, GO (2.0 g), urea (2.0 g, Sinopharm Chemical Reagent Co.), and $Fe(NO_3)_3 \cdot 9H_2O$ (2.4 g, Damao Chemical Reagent Co.) were dissolved in a mixture of ethylene glycol (40 mL, Hengxing Chemical Preparation Co.) and deionized water (290 mL) and then stirred and ultra-sounded for 2 h. The obtained liquid was transferred into a Teflon-lined stainless-steel autoclave, followed by one-pot hydrothermal synthesis at 180 °C for 12 h with rotation. The products were washed and filtered until neutral, then frozen to dry before calcining at 500 °C in a nitrogen atmosphere for 4 h (unless otherwise stated, all drying methods employed were freezing drying methods to maintain the graphene structure). Cobalt and potassium were loaded by impregnating $C_{10}H_{16}CoO_4$ (Macklin Biochemical Co.) and $K_2CO_3$ (Damao Chemical Reagent Co.) as cobalt and potassium sources, respectively. The amount of the impregnated material was calculated according to the given contents. The obtained catalyst was dried and calcined at 500 °C in a nitrogen atmosphere for 4 h. The as-prepared catalyst was designated as GO-Fe/K-Co. It should be noted that the portion before the slash represented the elements loaded by the one-pot hydrothermal method, while the portion after the slash represented the elements loaded by the impregnation method.

GO/K-Fe-Co was prepared in the following steps: Hydrothermal treatment of GO and urea was first performed, followed by impregnating $K_2CO_3$, $Fe(NO_3)_3 \cdot 9H_2O$, and $C_{10}H_{16}CoO_4$ as K, Fe, and Co sources onto the calcined catalyst. The drying and calcination processes after the hydrothermal synthesis and the impregnation remained unchanged.

In the same way, GO-Co/K-Fe and GO-Fe-Co/K catalysts were synthesized with the given loadings by changing the orders of material addition via the hydrothermal or impregnation methods without altering any other preparation techniques or stages.

To examine the influence of Fe contents, GO-Fe/K-Co with target Fe loadings of 25% and 30% were prepared and denoted as GO-25Fe/K-Co and GO-30Fe/K-Co, respectively. In addition, GO/K-10Fe-20Co was synthesized by loading 10% Fe and 20% Co with an impregnation process. Unlike the GO/K-Fe-Co catalyst mentioned above, Fe here was first impregnated onto the graphene surface, and then Co was loaded.

**Graphene-supported Fe catalysts.** GO-Fe/K and GO/K-Fe catalysts without the Co addition were also fabricated, in which the loadings for Fe and K were 20 wt% and 1 wt%, respectively.

**Other carbon materials supported Fe-Co bimetallic catalysts.** By adding the same amounts of $K_2CO_3$, $Fe(NO_3)_3 \cdot 9H_2O$, and $C_{15}H_{21}CoO_6$ to those of the GO-Fe/K-Co catalyst for comparison, carbon nanotubes (CNTs, Macklin Biochemical Co.), carbon black (CB, Macklin Biochemical Co.), and activated carbon (AC, Damao Chemical Co.) were utilized as supports for the preparation of Fe-Co bimetallic catalysts. These catalysts were designated as CNTs-Fe/K-Co, CB-Fe/K-Co, and AC-Fe/K-Co, respectively.

**rGO.** Only GO (2.0 g) and urea (2.0 g) were dissolved in the mixture of ethylene glycol (40 mL) and deionized water (290 mL), agitated, and transferred into a Teflon-lined stainless steel autoclave, where a

one-pot hydrothermal synthesis was carried out for 12 h at 180 °C. The resulting material was washed, dried, and calcined, the resultant graphene was labeled as rGO.

**rGO-Fe/K-Co**. The same synthetic process as for GO-Fe/K-Co was applied to the synthesis of rGO-Fe/K-Co, but the raw ingredient was rGO.

## Characterization

The actual total loadings of Fe, Co, and K in different catalysts were established using ICP-OES, which was performed on an Agilent 5110 (OES). The test procedures were as follows: a 10 mg sample was dissolved in a mix solution (2 mL $HNO_3$ + 6 mL $HCl$+2 mL $HF$) overnight. The dissolved sample was then added to a flask and diluted to the scale line. Five internal concentration standard solutions (0.5 mg/L, 1 mg/L, 3 mg/L, 5 mg/L, and 10 mg/L) were analyzed, and a standard curve was formed.

To acquire diffraction patterns, an X-ray diffractometer (XRD) with Cu Kα radiation was employed over a Rigaku RINT 2400 instrument (Scan angle: 5°–90°; scan speed: 2°/min; voltage and current: 40 kV and 40 mA). In situ XRD measurement was conducted on a SmartLab-TD diffraction system using a Cu Kα source with an XRK 900 heater. The reduction was carried out under the conditions of pure hydrogen and a temperature range of 25–400 °C. The carbonization process was performed on the reaction gas ($CO_2$/$H_2$) at 320 °C. Surface morphologies of the catalysts were examined using scanning electron microscopy (SEM, JEOL JSM-IT700HR), and transmission electron microscopy (TEM, JEOL JEM-2100F) was utilized to observe the morphologies and elemental mapping of the catalysts at an acceleration voltage of 100 kV. FIB (Focused ion beam)-SEM was performed using a double-beam electron microscope (Helios G4 PFIB CXe). The surface area of the catalysts was determined using $N_2$ adsorption-desorption experiments at −196 °C (Micromeritics 3Flex ASAP 2460). Prior to the tests, the samples were vacuum-degassed for 8 h at 240 °C. The M-H properties of the fresh catalysts were measured with a vibrating sample magnetometer (VSM, LakeShore 7404).

$H_2$ temperature-programmed reduction ($H_2$-TPR) tests were performed using a BELCAT-II-T-SP analyzer with a thermal conductivity detector (TCD). Helium was used as a pretreatment gas for the sample of 30 mg for 1 h at 300 °C. A gas mixture (5% $H_2$/Ar) was then delivered to the reactor at a rate of 30 mL/min when the temperature was decreased to 50 °C. Finally, $H_2$-TPR curves were obtained at temperatures ranging from 50 to 900 °C with a heating rate of 10 °C per minute. $CO_2$ or $H_2$ temperature-programmed desorption (TPD) tests were also investigated using the same apparatus. 30 mg of the sample was reduced for 2 h at 400 °C under a 100% $H_2$ gas flow (30 mL/min). The temperature of the reactor was reduced to 50 °C under a He gas flow (30 mL/min) after reduction. The reactor was subsequently filled with a 100% $CO_2$ or 5% $H_2$/Ar gas mixture for 1 h. He gas was then introduced into the reactor to remove the physically-adsorbed $CO_2$ or $H_2$. The $CO_2$-TPD and $H_2$-TPD curves were recorded from 50 to 900 °C with a heating rate of 10 °C per minute. $C_3H_6$-pulse transient hydrogenation experiments were performed on the spent catalysts. Before experiments, the spent catalysts were pretreated in pure $H_2$ at 350 °C for 2 h to activate the surface. And then the system was cooled to 320 °C in the Ar stream. After that, the samples were exposed to pure $H_2$. As the 10% $C_3H_6$/90% Ar gas was pulsed into the reactor, $CH_4$ ($m/z$ = 16), $C_3H_6$ ($m/z$ = 42), and $C_3H_8$ ($m/z$ = 44) transient signals were detected by a mass spectrometer.

For the X-ray photoelectron spectroscopy (XPS) analyses, an X-ray photoelectron spectrometer (KRATOS, Axis Ultra DLD) instrument was utilized, equipped with a catalyst pretreatment chamber for altering the gas composition. The excitation source was Al Kα ray (hν =1486.6 eV).

The $^{57}Fe$ Mössbauer spectra were recorded on an SEE Co W304 Mössbauer spectrometer, using a $^{57}Co$/Rh source in transmission geometry. The data were fitted using the MossWinn 4.0 software. Fourier transform infrared spectroscopy (FTIR) was conducted on a Thermo Scientific Nicolet iS20 IR spectrometer. The samples were finely milled, evenly combined with KBr, and pelletized. The spectral resolution was 4 $cm^{-1}$, and 32 scans were recorded for each spectrum. The Raman spectra were recorded at room temperature on a HORIBA Scientific LabRAM HR Evolution Raman spectrometer.

Co K-edge analyses were carried out with Si (111) crystal monochromators at the BL11B beamlines at the Shanghai Synchrotron Radiation Facility (SSRF) (Shanghai, China). Before the examination at the beamline, samples were compressed into thin sheets of 1 cm in diameter and sealed with Kapton tape film. The EXAFS spectra were captured using a 4-channel Silicon Drift Detector (SDD) Bruker 5040 at room temperature. The Co K-edge extended X-ray absorption fine structure (EXAFS) spectra were recorded in the transmission mode. Two scans were conducted for each sample, and negligible changes in the line shape and peak position of the Co K-edge XANES spectra were observed between the two scans. The EXAFS spectra of these standard samples (Co, CoO, and $Co_3O_4$) were also recorded in the transmission mode. The spectra were processed and analyzed using the software codes Athena and Artemis.

## Catalyst tests

Granular catalysts of 0.12 g (20–40 meshes) mixed with quartz sand of 0.5 g were used to evaluate the catalytic performance in a fixed-bed reactor. On top of the catalyst bed, 1 g glass beads are applied to adjust the bed height and preheat the reaction gas. The two ends of the catalyst bed and glass beads were separated by quartz cotton. Prior to the reaction, the catalyst was reduced for 8 h at 400 °C with pure $H_2$ of 30 mL/min. After reduction, the reactor was cooled to room temperature. The reactor was then filled with $CO_2$/$H_2$/Ar (27.0/68.0/5.0) reactant gas, and the temperature and pressure of the system were gradually raised to 320 °C and 3.0 MPa, respectively, and $W_{cat}$/$F_{CO2+H2}$ was 4.5 g h $mol^{-1}$.

To collect the heavy hydrocarbons and eliminate the water generated by the reaction, an ice trap was placed between the reactor and the back pressure valve, and an octane of 2 g was added to the ice trap in order to absorb heavy hydrocarbons. At the end of the reaction, the product in the ice trap was collected, and dodecane of 0.1 g and 2-butanol of 0.1 g were added as internal standards to the oil and water phases, respectively. An off-line gas chromatograph (Shimadzu GC-2014) equipped with a flame ionization detector (FID) and a DB-1 capillary column was used to examine the heavy hydrocarbons and water phase product. Two online gas chromatography systems (GL Sciences GC320 and Shimadzu GC-2014) were used to identify the gas-phase products: one had a thermal conductive detector (TCD, GC320) and an active charcoal column for analyzing Ar, CO, $CH_4$, and $CO_2$, while the other had an FID (GC-2014) and a GS-ALUMINA capillary column for analyzing light hydrocarbons.

## Statistics and reproducibility

We repeated the main catalysts for the catalyst test. All the experimental results can be reproduced within a small margin of error. No statistical method was used to predetermine the sample size. No data were excluded from the analyses. The experiments were not randomized. The investigators were not blinded to allocation during experiments and outcome assessment.

## Data availability

The source data generated in this study are provided in the Source Data file. Source data are provided with this paper.

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

## Acknowledgements

This work was supported by National Natural Science Foundation of China (22102001) (L.G.), NEDO (New energy and industrial Technology Development Organization) (JPNP16002) (N.T.), JST SPRING (JPMJSP2145) of Japan (J.M.L.), Liaoning province unveils science and technology project (2021JH1/10400101) (B.L.), and the Grant-in-Aid from Japan Society for the Promotion of Science (JSPS) (22H01864, 23H05404) (N.T.). We thank Bowei Meng and Hengyang Liu for synthesizing the catalysts and doing characterization during the revision process.

## Author contributions

J.M.L. and J.L. completed the catalyst tests and analyzed the data. L.G. wrote the paper with input from all the authors. W.W. and C.W. synthesized the catalysts. W.G. and X.G. did the XRD analysis. Y.H. did the catalyst morphology characterizations. G.Y. and S.Y. analyzed the characterization results. B.L. and N.T. revised the paper. All the authors contributed to the discussions on the results.

## Competing interests

The authors declare no competing interests.
