## [Peer Review File · Nature Communications]

Product-switchable CO₂ hydrogenation over Fe-Co bimetallic catalysts through a graphene fencing approachREVIEWER COMMENTS

Reviewer #1 (Remarks to the Author):

The manuscript of “Graphene Fence Engineering of Multiple Catalytic Sites Realizes Graphene Fence Engineering of Multiple Catalytic Sites Realizes” investigated the effect of addition order on the product selectivity in CO₂ hydrogenation reaction. The authors suggested that the graphene support acting as the fence can adjust the spatial distribution of multiple sites, which would switch the products selectivity from LPG to lower alkenes. I think the overall quality of the manuscript does not meet the high standard of Nature Communications. The reasons are as follows:

1. FeCo catalysts have been widely studied in recent works (Applied Catalysis B: Environmental, 2023, 328, 122476; ACS Catal. 2020, 10, 8660–8671; The Innovation, 2021, 2, 100170; ACS Catal. 2021, 11, 2267–2278; ACS Catal. 2016, 6, 913–927) to yield alkenes and liquid fuel and the effect of the spatial distribution have been investigated in detail in these previous articles. The novelty of this work is limited.
2. The role of graphene in this work is not clear due to the lacking of universality. From Figure 1, the signals of Fe and Co elements were overlapped in some regions. Hence, the role of graphene as fence is not clear. What about the other 2D materials?
3. The addition order of K is garbled. As the K promoter is crucial in CO₂ hydrogenation, the addition of K with Co or Fe also influence the performance.
4. In the catalyst preparation, Fe salt is Fe(NO₃)₃ while Co salt is C₁₀H₁₆CoO₄. Why did the authors choose organic salt for cobalt? After the calcination at 500 °C in a nitrogen atmosphere due to the GO, is there any residual carbon from C₁₀H₁₆CoO₄?
5. In the catalyst preparation of rGO-Fe/K-Co, rGO was fabricated via the a one-pot hydrothermal synthesis as GO without Fe. How to add the Fe species?

Reviewer #2 (Remarks to the Author):

In this manuscript, based on the proposed concept of “graphene fence engineering”, the authors reported a unique reaction system for manipulating the Fe-Co active sites, which enabled the switching of CO₂ hydrogenation products between light olefins and LPG. Among them, the catalyst for LPG simultaneously satisfied the demand for carbon growth and olefin secondary hydrogenation by separating the Fe and Co sites, thus obtaining a record-breaking STY through the Fischer-Tropsch pathway. The original work seems to have been well conducted with sufficient details, and I believe that the innovation of this manuscript can attract wide attention from the readers. Overall, I would recommend publication in Nature Communications with several suggestions below.

1. In Fig.1b, the authors found that after the hydrothermal, the graphene diffraction peak shifted to a higher position, indicating a decrease in graphene layer spacing. In the reduction process, does the graphene peak gradually move to the higher direction, and when the peak is about 25°, is it completely reduced?

2. According to the TEM images, both GO-Fe-Co/K and GO/K-Fe-Co exhibited smaller particle sizes. However, the iron carbide contents of these two spent catalysts showed significant differences in the ^{57}Fe Mössbauer spectra. Please explain this phenomenon. In addition, is the high light olefin selectivity of GO/K-Fe-Co caused by the iron carbides?

3. In the catalytic performance part, the authors attributed the poor performance of GO-Co/K-Fe to excessive carbon deposition, which is explained by the lower content of Fe and Co in TEM mapping (line 336, page 17). However, this may only be related to the TEM region selection leading to the higher carbon content.

4. For the GO-Fe/K-Co catalyst, the detailed product distribution should be provided to explore the olefin secondary hydrogenation.

5. By connecting the Fe and Co active sites, the authors realized the reaction path of carbon chain growth and then olefin hydrogenation, thus, the products were concentrated in propane and butane. Have the authors tried to physically mix the hydrogenation catalyst with the carbon chain growth catalyst macroscopically to explore the product selectivity?

6. The authors calculated the energy barriers by DFT calculations and concluded that the olefin secondary hydrogenation reaction on the $\text{Fe}_5\text{C}_2/\text{Co}$ sites was easier to happen than that on the Fe_5C_2 sites. Moreover, it also needs to be proven that the Fe_5C_2 site is easier to carry out the carbon chain growth reaction.

Reviewer #3 (Remarks to the Author):

In this paper, the authors presented a Graphene Fence Engineering of Multiple Catalytic Sites for Switchable-Orientation CO_2 Hydrogenation. They designed Fe-Co nanoparticles over graphene fence leading to catalytic process for the direct conversion of CO_2/H_2 mixtures into different types of hydrocarbons. The spatial dual sites of Fe-Co nanoparticles separated by graphene fences achieved 43.6% LPG (C3P-C4P) selectivity and the low CO production of 2.2% at CO_2 conversion of 46%, while Fe-Co with an intimate contact being anchored on the surface of graphene fences presented 50.1% light olefin (C2=C4=) selectivity at a CO_2 conversion of 55.4%. With the assistance of graphene fences, the synergy between iron carbides and metallic cobalt could efficiently regulate olefin secondary hydrogenation, achieving a selective switch from light olefins to LPG for CO_2 hydrogenation. The catalytic results are very exciting and the if the principle works, it can be a breakthrough in Power to X PTX industry, which is a crucial platform technology for future net zero economy. Various catalysts samples like GO-Fe/K-Co and GO/K-Fe-Co and nanotube supported Co-Fe have been prepared and lots of interesting results are obtained, in general, it is shown that graphene fence plays a big role in the process and the metal loading ways also has big influence. In general it is a nice paper,

and can be accepted for publication after the following being addressed:

1. The catalyst test section, I can not find how much catalyst samples were used for the test, and information on the catalyst physical properties. Given the unique performance of the catalysts, physical properties like shape, particle size play key roles in the reactants and products penetration and diffusion.
2. Why $C_{15}H_{21}CoO_6$ was used for Co source? Can $Co(NO_3)_2$ make any difference?
3. How about the temperature effect on the product distribution and CO_2 conversion? This may be relevant to the reaction mechanism.
4. How about the metal and graphene changes in the catalysts after the activity tests? Ideally the characterization data on the working catalysts at various stage would be great help for understanding the catalyst and reaction.

Point-by-point responses to the comments (NCOMMS-23-31318)

We are very grateful to the editor and referees for the review and valuable suggestions on the manuscript. The following descriptions are the responses to these comments, one by one. The questions from reviewers and the actions were highlighted in light blue words and yellow backgrounds, respectively, and the changes made during the revision were highlighted in red words in the revised manuscript and supporting information file.

Reviewer #1 (Remarks to the Author):

Overall Comments: The manuscript of “Graphene Fence Engineering of Multiple Catalytic Sites Realizes Graphene Fence Engineering of Multiple Catalytic Sites Realizes” investigated the effect of addition order on the product selectivity in CO₂ hydrogenation reaction. The authors suggested that the graphene support acting as the fence can adjust the spatial distribution of multiple sites, which would switch the products selectivity from LPG to lower alkenes. I think the overall quality of the manuscript does not meet the high standard of Nature Communications. The reasons are as follows:

Responses:

We thank Referee #1 for the careful review. In our manuscript entitled “Graphene fence engineering of multiple catalytic sites realizes switchable-orientation CO₂ hydrogenation”, the concept of “graphene fence” was first proposed in C1 chemistry, and a series of Fe-Co bimetallic catalysts with different metal distributions were precisely constructed by utilizing the GO structural transformation during the reduction process. Due to the different properties of Fe and Co in catalyzing CO₂ hydrogenation reactions, different distributions of Fe-Co active sites can achieve the effect of product selective switching between light olefins (C₂⁼–C₄⁼) and

LPG ($C_3^P-C_4^P$) without changing the total metal loading content. Both the designed catalysts exhibited ultra-high catalytic activity and target product selectivity (**Scheme R1**). In order to facilitate readers understanding our work idea, the **Scheme R1** was added as a new **Supplementary Scheme 1** in the revised supporting information file. We hope that the reviewer finds our responses satisfactory. The responses to comments are listed below.

Scheme R1 | Schematic diagram of graphene fences regulating metal distributions and product types. Fe species, red. Co species, blue. K species, purple.

Specific Comments:

Comment 1. FeCo catalysts have been widely studied in recent works (Applied Catalysis B: Environmental, 2023, 328, 122476; ACS Catal. 2020, 10, 8660–8671; The Innovation, 2021, 2, 100170; ACS Catal. 2021, 11, 2267–2278; ACS Catal. 2016, 6, 913–927) to yield alkenes and liquid fuel and the effect of the spatial distribution have been investigated in detail in these previous articles. The novelty of this work is limited.

Responses:

We thank Referee #1 for the concerns. More recently, bimetallic catalysts, especially Fe-Co alloys, have been extensively studied, as shown by the works cited by Referee #1. However,

almost all of these articles are devoted to the study of active phases. For example, in these works, the researchers found that the χ -($\text{Co}_x\text{Fe}_{1-x}$) $_5\text{C}_2$ alloy carbides have a stronger carbon chain growth capacity relative to the cobalt carbide phases or iron carbide phases, so the catalysts with more Fe-Co alloy carbide phases exhibited higher selectivity for light olefins or long-chain olefins. These works just focus on the single active phase (($\text{Co}_x\text{Fe}_{1-x}$) $_5\text{C}_2$), which plays a single role in carbon chain growth. By adjusting the content of active metals or promoters, they got the most of the ($\text{Co}_x\text{Fe}_{1-x}$) $_5\text{C}_2$ phase content (**Scheme R2**). But in our work, under the premise of keeping the total metal content unchanged, by regulating the spatial distributions of Fe and Co sites, such as intimate contact or spatially scattered sites via tailor-made graphene fabrication, we can achieve the selective switching of product types through different synergy effects on different sites and obtain pretty product distributions (**Scheme R2**). We think this work departs significantly from previous studies of Fe-Co bimetallic catalysts, and the innovation justifies publication in *Nature Communications*. In addition, the novelty of this manuscript is also reflected in the following aspects (I–III):

Scheme R2 | Differences between this work and previous work. Fe species, red. Co species, blue. K species, purple.

I. Utilize graphene as self-forming fences to separate metal active sites.

In past research, graphene was always used as support to merely anchor or disperse metal sites for better stability or enhancing catalytic performances (*Angew. Chem. Int. Ed.* 2023, 135, e202217071; *ACS Catal.* 2016, 6, 389–399; *ACS Appl. Mater. Interfaces.* 2018, 10, 23439–23443; *Catal Today.* 2020, 355, 10–16). However, in this work, we creatively introduced the bimetallic system to the graphene support and successfully prepared bimetallic catalysts with varied but suitable spatial distributions by utilizing the structural transformation of graphene (Scheme R3).

Scheme R3 | Schematic diagram of graphene fences dividing metal active sites. Fe species, red. Co species, light blue. K species, purple.

II. Switch the product types without changing the metal content.

Tuning CO₂ hydrogenation selectivity to obtain target products with high selectivity has attracted increasing attention. The selective control of catalytic products is the core of catalysis science, and it is also the difficulty of catalysis research. However, the majority of previous relevant studies stay at the level of adjusting the target product by changing the metal content or interaction (*Angew. Chem. Int. Ed.* 2020, 59, 19983–19989). The discovery of selective

catalysis is of great scientific value. For example, recently, Bao group modulated the interaction of Co and Zn to switch the product between CO and CH₄ by changing the mixing method on a macro level (*J. Am. Chem. Soc.* 2023, 145, 17056–17065). Our work convincingly manipulated the spatial distributions of active sites at the nanoscale and enabled the product to switch between higher-value-added light olefins and LPG (**Scheme R1**). As a result, this work is worthy of publication in the journal at this level.

III. Create a precedent for CO₂ hydrogenation to LPG via a Fischer-Tropsch pathway.

To our knowledge, almost all the LPG synthesis methods, from CO₂ hydrogenation until now, employed a methanol-intermediated route by combining methanol synthesis catalysts with zeolites. No related Fischer-Tropsch route was reported for selective LPG synthesis from CO₂ hydrogenation. Our DFT calculation results revealed the difficulty of producing LPG via a Fischer-Tropsch pathway, that is, the contradiction between the carbon chain growth and the olefin secondary hydrogenation (**Fig. R1**). For active sites with weak hydrogen adsorption capacity, such as Fe₅C₂, it is difficult for alkenes to be hydrogenated to alkanes, leading to low alkane selectivity. Whereas for the sites with strong hydrogen adsorption capacity, such as Fe₅C₂-Co, on the other hand, it is also difficult to achieve carbon growth, and excessive methane and ethane products reduce the LPG selectivity. In this situation, the proposed graphene-fence-separated dual active sites could simultaneously meet the demands of carbon chain growth and olefin hydrogenation, thus overcoming this difficulty. As expected, the newly proposed Fischer-Tropsch path exhibited an ultra-high STY_{LPG} (space-time yield) (151.0 g kg_{cat}⁻¹ h⁻¹), which was much higher than any other previously reported composite methanol-intermediate catalysts (*including Nature Catalysis* 2022, 5, 1038–1050) [1–6], so as to better meet the needs of industrial production (**Fig. R2**).

Thus, we think this manuscript is innovative compared to the previous related work. As Referee #3 evaluated, “The catalytic results are very exciting, and if the principle works, it can be a breakthrough in the Power to X PTX industry, which is a crucial platform technology for future net zero economy.” As a result, my co-authors and I believe that the novelty of our work

is suitable for publication in *Nature Communications*. We hope the reviewer is satisfied with our revision.

In order to give the readers a better understanding of the respective functions and synergy effects of the two active sites, we added DFT calculations as a new **Fig. 5** in the revised manuscript, and the descriptions were added to **Pages 26 and 27** in red words.

Fig. R1 | The new Fischer-Tropsch pathway for CO₂ hydrogenation to LPG. (a) Synergistic effect of dual active sites for producing LPG from CO₂ hydrogenation. The numbers represent the free energy barrier (eV). (b–c) DFT calculations of chain growth and olefin hydrogenation reactions from ethylene to propane over the Fe₅C₂ site and the Fe₅C₂-Co site. Red and blue balls represent Fe and Co species, respectively.

Fig. R2 | STY_{LPG} of GO-Fe/K-Co compared with other previously reported catalysts. Catalysts from left to right were cited from the references [1–6] (see the end of the Responses file).

Comment 2. The role of graphene in this work is not clear due to the lacking of universality. From Figure 1, the signals of Fe and Co elements were overlapped in some regions. Hence, the role of graphene as fence is not clear. What about the other 2D materials?

Responses:

We thank Referee #1 for the doubts arising from careful consideration. As a widely used support material, graphene has a representative feature, that is, its structure would change in the reduction process, from a two-dimensional lamellar structure to a three-dimensional structure (*Catal Today*. 2020, 355, 10–16; *J. Electron. Spectrosc.* 2014, 195, 145–154). Using the encapsulating effect of the graphene fences during the graphene structural evolution, we successfully synthesized a series of catalysts with different internal and external metal distributions by changing the introduction orders of Fe and Co. These catalysts exhibited significantly different catalytic performances. In particular, the product selective switching between light olefins and LPG (propane and butane) could be achieved by manipulating Fe and Co sites.

Indeed, due to the large specific surface area of graphene, partial iron was loaded onto the external layers and could not be encapsulated into the graphene fences, forming Fe-Co active

sites with Co introduced in the subsequent impregnation process (**Scheme R4**). These Fe-Co sites caused the overlap of Fe and Co signals in TEM mapping (**Fig. 1f**).

However, due to the electron transfers from Co to Fe₅C₂ after Co was introduced (**Fig. R3a**) and the electron-deficient environment is more conducive to hydrogen adsorption (*Ind. Eng. Chem. Res.* 2019, 58, 21350–21362), even if graphene fences cannot completely separate the Fe and Co, partial Fe₅C₂ sites encapsulated in the inner layers can still form different strengths of hydrogen adsorption with the Fe₅C₂-Co sites on the external layers (**Fig. R3b**).

The DFT calculations also showed that the Fe₅C₂-Co site with stronger hydrogen adsorption capacity is more inclined to hydrogenate alkenes to alkanes, while the Fe₅C₂ site with weaker hydrogen adsorption capacity is more inclined to facilitate the carbon chain growth reaction (**Fig. R4**). Combined with catalyst test results, the synergistic effect of the dual active sites can simultaneously meet the demands of carbon chain growth and olefin secondary hydrogenation. As a result, the products were concentrated in LPG (43.6%).

We also employed other carbon materials as support for Fe-Co bimetallic catalysts, such as active carbon (AC), carbon black (CB), and CNTS. AC-Fe/K-Co, CB-Fe/K-Co, and CNTS-Fe/K-Co catalysts were prepared using the same synthesis method as GO-Fe/K-Co: the support was first hydrothermal synthesis with Fe and then impregnated with Co and K. TEM mapping distributions of these three catalysts were tested, and the results were shown in **Fig. R5**. As observed, Co distributions were found in all the regions with Fe distributions. Some areas where only Co was distributed could be attributed to the uneven dispersion of Co during the latter impregnation process. Therefore, these three carbon materials (AC, CB, and CNTS) did not exhibit the same expected encapsulation and fence effects as graphene.

We added the **Fig. R3** and **Fig. R5** as new **Supplementary Fig. 26** and **Supplementary Fig. 23**, respectively, in the revised supporting information file and the descriptions were added to **Pages 22 and 19**, respectively, in red words in the revised manuscript.

Scheme R4 | Schematic diagram of the formation of Fe-Co sites and Fe sites. Fe species, red. Co species, blue. K species, purple.

Fig. R3 | Charge density difference of $\text{Fe}_5\text{C}_2\text{-Co}$ site (a) and Hydrogen adsorption energies of $\text{Fe}_5\text{C}_2\text{-Co}$ site and Fe_5C_2 sites (b). The red and green colors represented electron accumulation and loss, respectively. Element colors: Fe, pink. Co, light blue. C, grey. H, white.

Fig. R4 | The new Fischer-Tropsch pathway for CO₂ hydrogenation to LPG. (a) Synergistic effect of dual active sites for producing LPG from CO₂ hydrogenation. The numbers represent the free energy barrier (eV). (b–c) DFT calculations of chain growth and olefin hydrogenation reactions over the Fe₅C₂ site and the Fe₅C₂-Co site. Red and blue balls represent Fe and Co species, respectively.

Fig. R5 | TEM mapping distributions of Fe and Co. (a) AC-Fe/K-Co. (b) CB-Fe/K-Co. (c) CNTS-Fe/K-Co.

Comment 3. The addition order of K is garbled. As the K promoter is crucial in CO₂ hydrogenation, the addition of K with Co or Fe also influence the performance.

Responses:

We would like to thank Referee #1 for the insightful perspective. The potassium content is relatively small compared to iron and cobalt, and K is involved in the reaction as a promoter rather than a main active component, like Fe and Co. As a result, considering the complexity of our catalyst system, we did not change the loading method of K as Fe and Co, but added K by impregnation method in all the catalysts. Our experimental design idea is as follows:

Previous studies have shown that the addition of potassium can promote the formation of olefins and inhibit the formation of paraffins by facilitating electron transfer from potassium to iron species (*ACS Catal.* 2020, 10, 12098–12108; *ACS Catal.* 2020, 10, 14516–14526; *Chem*

Eng Sci. 2023, 282, 119228). Therefore, in the olefin-producing catalysts, such as GO/K-Fe-Co, K should be loaded to the Fe-Co active sites as much as possible, while in the LPG-producing catalysts (GO-Fe/K-Co), the addition of K should be avoided to reduce the LPG selectivity. Since most of the Fe and Co particles in the olefin-producing catalyst (GO/K-Fe-Co) were supported on the external layers of graphene, we chose to load K in the same way of impregnation in all the catalysts.

The metal distributions of GO/K-Fe-Co and GO-Fe/K-Co are shown in **Fig. R6**. As expected, in GO/K-Fe-Co, K and Fe-Co exhibited the same distributions due to the well dispersion, whereas in GO-Fe/K-Co, K and Fe-Co showed different distributions due to the agglomeration of metal particles and the separation effect of the graphene fences. Thus, little K was adjacent to Fe. Correspondingly, the addition of K facilitated the light olefin formation (50.1%) in GO/K-Fe-Co, while not changing the product type in GO-Fe/K-Co because only little K was adjacent to the Fe-Co particles and inhibited the hydrogen adsorption.

Fig. R6 | TEM mapping distributions of GO/K-Fe-Co and GO-Fe/K-Co.

Comment 4. In the catalyst preparation, Fe salt is $\text{Fe}(\text{NO}_3)_3$ while Co salt is $\text{C}_{10}\text{H}_{16}\text{CoO}_4$. Why did the authors choose organic salt for cobalt? After the calcination at 500 °C in a nitrogen atmosphere due to the GO, is there any residual carbon from $\text{C}_{10}\text{H}_{16}\text{CoO}_4$?

Responses:

We would like to thank Referee #1 for the meticulous observation of our experiment details. As is known to all, the hydrothermal synthesis process may cause the loss of catalysts, and the target Co contents in our catalysts (4%) are far less than the target Fe contents (20%). As a result, the Co loss in the hydrothermal process may bring huge errors to the final Co content and make it difficult to be manipulated. Therefore, we conducted a preliminary screening of the varied Co sources before the experiment and found that cobalt acetylacetonate was more easily anchored to the graphene support than cobalt nitrate, as shown in **Table R1**. The same weight of GO was hydrothermally synthesized with cobalt acetylacetonate or cobalt nitrate containing the same molar amount of Co, respectively. The products using cobalt acetylacetonate as Co sources exhibited a higher Co content, which was obtained by ICP-OES tests, possibly due to the affinity between organic groups on graphene surface and cobalt acetylacetonate. Meanwhile, cobalt-based catalysts with cobalt acetylacetonate as sources have been proven to be well dispersed on graphene (*Catal Today*. 2020, 355, 10–16). As a result, we chose cobalt acetylacetonate as Co sources for hydrothermal synthesis. Furthermore, to control the variables, cobalt acetylacetonate was used as the Co source in both hydrothermal and impregnation processes.

Table R1 | Hydrothermal synthesis results of GO with different Co sources.

Co sources	Co molar of Co sources	Weight of Co sources	GO weight	Co content ^a
$C_{10}H_{16}CoO_4$	0.002 mol	0.518 g	2 g	5.43%
$Co(NO_3)_2 \cdot 6H_2O$	0.002 mol	0.582 g	2 g	5.21%

^a Co content was obtained by ICP-OES tests.

In order to investigate the decomposition process of cobalt acetylacetonate, we applied TG-DTG tests in the air atmosphere and N_2 atmosphere, as shown in **Fig. R7**. As the temperature rose at a rate of 5 °C/min, the sample quantity gradually decreased. In the air atmosphere, when the temperature reached 376 °C, the sample quantity no longer decreased even at 800 °C, while in the N_2 atmosphere, the sample weight reached stability at about 493 °C, staying stable until 800 °C. As a result, the organic carbon in cobalt acetylacetonate could be completely decomposed in a N_2 atmosphere at 500 °C for 4 hours.

Fig. R7 | Thermal analysis curve of cobalt acetylacetonate in air and N₂ atmosphere.

Comment 5. In the catalyst preparation of rGO-Fe/K-Co, rGO was fabricated via a one-pot hydrothermal synthesis as GO without Fe. How to add the Fe species?

Responses:

We apologize that our expression made Referee #1 confused. Please let us explain this synthesis process in detail.

First, please allow us to clarify our rules for naming catalysts once again. For example, GO-Fe/K-Co means that we employed graphene oxide and iron sources as raw materials for hydrothermal synthesis and then used K and Co sources for impregnation, in which of the nomenclature, the part before the slash line represents the raw materials for hydrothermal synthesis and the part after the slash line represents the raw materials for impregnation. **Scheme R5** intuitively illustrated this naming convention.

As a result, rGO-Fe/K-Co means that we used rGO (reduced graphene oxide) instead of GO for the hydrothermal synthesis with ferric nitrate. At this time, Fe was introduced. Before this process, rGO was obtained via a one-pot hydrothermal synthesis of GO without Fe. The synthesized processes of GO-Fe/K-Co and rGO-Fe/K-Co are shown in **Scheme R6**.

In other words, in GO-Fe/K-Co, the reduction of graphene oxide was completed in the process of hydrothermal synthesis together with Fe, while in rGO-Fe/K-Co, the GO was first reduced before conducting hydrothermal synthesis again with Fe. Two catalysts exhibited

significantly different product selectivity, which indicated the effect of GO structural transformation on the metal active site distributions and further confirmed the role of graphene fences.

To make it easier for the readers to catch the content, the original paragraph is divided into two paragraphs and marked in red font, as shown in Page 33 in the catalyst preparation section.

Scheme R5 | Synthesis of Fe-Co bimetallic catalysts with different distributions.

Scheme R6 | Preparation processes of GO-Fe/K-Co and rGO-Fe/K-Co. (a), synthesis of rGO; (b), synthesis of GO-Fe/K-Co; (c), synthesis of rGO-Fe/K-Co.

In order to make it easier for readers to understand the catalyst naming method and

preparation processes, we added **Scheme R5** and **Scheme R6** as new **Supplementary Scheme 2** and **Supplementary Scheme 3**, respectively, in the revised supporting information file.

Reviewer #2 (Remarks to the Author):

Overall comment: In this manuscript, based on the proposed concept of “graphene fence engineering”, the authors reported a unique reaction system for manipulating the Fe-Co active sites, which enabled the switching of CO₂ hydrogenation products between light olefins and LPG. Among them, the catalyst for LPG simultaneously satisfied the demand for carbon growth and olefin secondary hydrogenation by separating the Fe and Co sites, thus obtaining a record-breaking STY through the Fischer-Tropsch pathway. The original work seems to have been well conducted with sufficient details, and I believe that the innovation of this manuscript can attract wide attention from the readers. Overall, I would recommend publication in *Nature Communications* with several suggestions below.

Responses:

We gratefully thank Referee #2 for the review and praise of our manuscript. We employed “graphene fences” to manipulate the distribution of Fe-Co active sites, thus switching the product types of CO₂ hydrogenation. The concept of “graphene fence engineering” proposed by our group provides a new idea for the application of graphene materials as catalyst support.

Furthermore, in response to the reviewer’s suggestions, we make the following improvements and explanations.

Specific Comments:

Comment 1. In Fig.1b, the authors found that after the hydrothermal, the graphene diffraction peak shifted to a higher position, indicating a decrease in graphene layer spacing. In the

reduction process, does the graphene peak gradually move to the higher direction, and when the peak is about 25°, is it completely reduced?

Responses:

We would like to express our gratitude to Referee #2 for the meaningful questions. We added the *in situ* XRD test of graphene oxide (GO) to detect the graphene characteristic peak shift during the temperature programmed reduction process, as shown in **Fig. R8**. The reduction was carried out under the conditions of pure hydrogen, and the temperature was increased from 25 °C to 400 °C at a rate of 3 °C per minute.

Initially, the graphene peak around 8° gradually shifted in a positive position. When the temperature rose to 205 °C, the second peak appeared at about 23° and the first peak disappeared. Subsequently, as the temperature rose, the second peak gradually shifted in a positive direction. However, as the reduction process continued, the graphene peak stayed at about 25° and no longer moved significantly. Thus, compared to XRD results of the fresh Fe-Co bimetallic catalysts (**Fig. 1a**), the graphene in the as-prepared catalysts was completely reduced.

We added **Fig. R8** as a new **Supplementary Fig. 2** in the revised supporting information file to show the movement trend of GO characteristic peaks during the reduction process, and the descriptions were added to **Page 7** in red words in the revised manuscript. The *in situ* XRD experimental details were also added to **Page 33** in red words in the revised manuscript.

Fig. R8 | *In situ* XRD for GO during temperature programmed reduction. Pure H₂, 30 mL/min, 25–400 °C, 3 °C/min, atmospheric pressure.

Comment 2. According to the TEM images, both GO-Fe-Co/K and GO/K-Fe-Co exhibited smaller particle sizes. However, the iron carbide contents of these two spent catalysts showed significant differences in the ⁵⁷Fe Mössbauer spectra. Please explain this phenomenon. In addition, is the high light olefin selectivity of GO/K-Fe-Co caused by the iron carbides?

Responses:

We would like to extend our gratitude to Referee #2 for the valuable suggestion. Indeed, the catalyst particle sizes affect the iron carbonization process. On the other hand, the interaction between metal and support is also critical for the formation of iron carbide, it has been proven that strong metal-support interaction (SMSI) inhibits the reduction of iron oxides and suppresses the carbonization (*Fuel*. 2022, 319, 123613; *Catal. Sci. Technol.* 2020, 10, 502–509). As proven by H₂-TPR profiles (**Supplementary Figure 19**), the GO-Fe-Co/K catalyst exhibited the highest temperature of reduction peak, indicating the strongest metal-support interaction and lowest reduction ability.

To intuitively illustrate this view, we compared the *in situ* XRD patterns on the reduction and carbonization processes of the as-prepared catalysts and showed it in **Fig. R9**. When the temperature rose to 400 °C, a peak of metallic iron appeared on the GO-Fe-Co/K catalyst.

Obviously, this temperature was the highest of all the catalysts. On the contrary, the GO/K-Fe-Co showed the lowest temperature (325 °C). Correspondingly, in the carbonization process, the Fe₅C₂ peak strength of GO/K-Fe-Co was significantly higher than that of GO-Fe-Co/K while the strength of the Fe peak was much lower than that of GO-Fe-Co/K (**Fig. R9**). As a result, due to its excessively strong metal-support interaction, the GO-Fe-Co/K catalyst exhibited a lower iron carbide content compared to the GO/K-Fe-Co.

Moreover, past research has illustrated that in the process of CO₂ hydrogenation, iron oxide is responsible for the RWGS reaction and iron carbide is responsible for the chain growth reaction (*Nat Commun.* 2020, 11, 6219; *ACS Catal.* 2022, 12, 7609–7621). So the GO/K-Fe-Co with a higher iron carbide content exhibited a higher light olefin selectivity compared to the GO-Fe-Co/K, whose products were mainly methane and ethane.

We added **Fig. R9** as a new **Supplementary Fig. 21** to explore the reduction and carbonization processes and the descriptions were added to **Page 17** in red words in the revised manuscript.

Fig. R9 | *In situ* XRD patterns for the reduction and carbonization processes of catalysts. Test conditions: Pure H₂ from 25–400 °C and reaction gas (CO₂/H₂) at 320 °C, atmospheric pressure,

30 mL/min. The left images represent the reduction process in a hydrogen atmosphere, and the right images represent the carbonization process in a reaction gas atmosphere.

Comment 3. In the catalytic performance part, the authors attributed the poor performance of GO-Co/K-Fe to excessive carbon deposition, which is explained by the lower content of Fe and Co in TEM mapping (line 336, page 17). However, this may only be related to the TEM region selection leading to the higher carbon content.

Responses:

We appreciate Referee #2 for pointing out this neglect. We apologize for any misunderstandings that may have arisen due to these misleading statements. Indeed, the lower content of Fe and Co may only be related to the TEM region selection. However, the lower Fe/Co ratio of the spent GO-Co/K-Fe catalyst can indicate excessive carbon deposition due to the amorphous carbon coating on the surface of the iron carbide affecting the detect of Fe content, especially compared to the fresh GO-Co/K-Fe catalyst (**Table R2 and Table R3**).

We added **Table R2** as a new **Supplementary Table 10** in the revised supporting information file and the paragraph in **Page 18** was corrected as follows in red words: **Clearly, as determined by TEM mapping, the ratio of Fe/Co in the spent GO-Co/K-Fe was much lower than that of other catalysts and the fresh GO-Co/K-Fe (Supplementary Fig. 22, Supplementary Table 10 and Supplementary Table 11), which can be explained as the large amount of amorphous carbon deposited on the surface of Fe sites affecting the determination of element contents.**

Table R2 | Element contents of the fresh GO-Co/K-Fe obtained by TEM mapping analysis.

Element	Line Type	k factor	Absorption Correction	wt%
C	K series	2.50675	1.00	51.22
K	K series	0.96973	1.00	1.19
Fe	K series	1.19079	1.00	38.90
Co	K series	1.26119	1.00	8.69

Table R3 | Element contents of spent catalysts obtained by TEM mapping analysis.

Catalysts	C (wt%)	Fe (wt%)	Co (wt%)	K (wt%)
GO-Co/K-Fe	95.9	1.6	1.8	0.7
GO/K-Fe-Co	56.4	39.6	2.9	1.1
GO-Fe-Co/K	82.0	14.2	3.3	0.5
GO-Fe/K-Co	59.4	35.7	4.4	0.5

Comment 4. For the GO-Fe/K-Co catalyst, the detailed product distribution should be provided to explore the olefin secondary hydrogenation.

Responses:

We appreciate Referee #2 bringing our attention to some critical details in our manuscript. For comparison, the detailed product distributions of GO-Fe/K and GO-Fe/K-Co catalysts have been added to **Fig. R10**. Before the Co addition, the product selectivity of C2 and C3 in GO-Fe/K was roughly the same. However, after the introduction of Co, C3 products occupied the highest selectivity, which further indicated that the diffusion and hydrogenation effects on the internal-active-site products shifted the chemical equilibrium in a positive direction, thus the products were concentrated in propane.

We have added **Fig. R10** as a new **Supplementary Fig. 14** in the revised supporting information file, and the descriptions were added to **Page 16** in red words in the revised manuscript.

Fig. R10 | Detailed product distributions of GO-Fe/K and GO-Fe/K-Co catalysts.

Comment 5. By connecting the Fe and Co active sites, the authors realized the reaction path of carbon chain growth and then olefin hydrogenation, thus, the products were concentrated in propane and butane. Have the authors tried to physically mix the hydrogenation catalyst with the carbon chain growth catalyst macroscopically to explore the product selectivity?

Responses:

We would like to thank Referee #2 for the insightful suggestions on our manuscript. We combined the catalyst with the highest methane selectivity (GO/K-10Fe-20Co) and the catalyst with the highest light olefin selectivity (GO/K-Fe-Co) together at a weight ratio of 1:1, and tested this composite catalyst under the same reaction conditions.

In terms of the results (Table R4), after combining the two catalysts together, C₂^P-C₄^P products were significantly increased. On the contrary, C₂⁼-C₄⁼ products were obviously decreased, demonstrating that alkenes were partially hydrogenated to alkanes. However, due to the long distance between different active sites caused by physically mixing, the efficiency of olefin secondary hydrogenation was limited, so that the alkenes more easily diffused into the gas phase and partial olefins were still produced.

Table R4 | Catalytic performances of GO-Fe/K-Co catalyst with different temperature.

Catalysts	CO ₂ conv. (%)	CO sel. (%)	Selectivity (%)					
			CH ₄	C ₂ ⁼ -C ₄ ⁼	C ₂ ^P	C ₃ ^P -C ₄ ^P	C ₄ ^{iso}	C ₅ ⁺
GO/K-10Fe-20Co	51.3	2.8	54.9	6.0	20.0	14.5	0.2	4.4
GO/K-Fe-Co	55.4	5.8	13	50.1	9.1	3.1	1.4	23.3
GO/K-10Fe-20Co & GO/K-Fe-Co	52.1	4.3	38.6	18.7	17.2	11.2	1.0	13.3

Reaction conditions: 320 °C, W/F=4.5 g h mol⁻¹ (relative to the total catalyst weight), 3.0 MPa, granule mixing at a weight ratio of 1:1, reduced in pure hydrogen for 8 hours before reaction.

Comment 6. The authors calculated the energy barriers by DFT calculations and concluded that the olefin secondary hydrogenation reaction on the Fe₅C₂/Co sites was easier to happen than that on the Fe₅C₂ sites. Moreover, it also needs to be proven that the Fe₅C₂ site is easier to carry out the carbon chain growth reaction.

Responses:

We would like to thank Referee #2 for the suggestions to make our manuscript more comprehensive. In order to determine the reaction difficulty of the carbon chain growth and olefin secondary hydrogenation processes, DFT calculations were employed to calculate the free energy barriers for the reactions from ethylene to butane, as summarized in **Fig. R11–14**. Over the Fe_5C_2 active site, the carbon chain growth reactions are easier to perform compared to the olefin secondary hydrogenation reactions due to their lower energy barriers (**Fig. R11b**). However, after Co is introduced, olefin secondary hydrogenation is more likely to be realized (**Fig. R11c**). These results further illustrate that the combination of the Fe_5C_2 and $\text{Fe}_5\text{C}_2\text{-Co}$ active sites can simultaneously meet the demand of carbon chain growth and olefin secondary hydrogenation, thus increasing the LPG (propane and butane) selectivity (**Fig. R11a**).

We have added **Fig. R11** as a new **Fig. 5** in the revised manuscript and **Fig. R12–14** as Supplementary **Fig. 32–34** in the revised supporting information file. Furthermore, the descriptions were added to **Pages 26 and 27** in red words in the revised manuscript.

Fig. R11 | The new Fischer-Tropsch pathway for CO₂ hydrogenation to LPG. (a) Synergistic effect of dual active sites for producing LPG from CO₂ hydrogenation. The numbers represent the free energy barrier (eV). (b–c) DFT calculations of chain growth and olefin hydrogenation reactions over the Fe₅C₂ site and the Fe₅C₂-Co site. Red and blue balls represent Fe and Co species, respectively.

Fig. R12 | Free energy barriers of ethylene coupled to butene over Fe₅C₂ and Fe₅C₂-Co site.

Note: Over the Fe₅C₂ site, the free energy barrier of ethylene coupling (1.68 eV) was higher than that of C₂H₄ growth to C₃H₆ (1.58 eV), while over the Fe₅C₂-Co site, the free energy barrier of ethylene coupling (2.10 eV) was higher than that of C₂H₄ hydrogenation to C₂H₆ (0.39 eV), indicating that the ethylene coupling pathway was not the ideal path over the dual active sites.

Fig. R13 | Optimized transition states and intermediates of various steps in carbon chain growth and olefin secondary hydrogenation reactions over Fe₅C₂ sites.

Fig. R14 | Optimized transition states and intermediates of various steps in carbon chain growth and olefin secondary hydrogenation reactions over $\text{Fe}_5\text{C}_2\text{-Co}$ sites.

Reviewer #3 (Remarks to the Author):

Overall Comment: In this paper, the authors presented a Graphene Fence Engineering of Multiple Catalytic Sites for Switchable-Orientation CO_2 Hydrogenation. They designed Fe-Co nanoparticles over graphene fence leading to catalytic process for the direct conversion of CO_2/H_2 mixtures into different types of hydrocarbons. The spatial dual sites of Fe-Co nanoparticles separated by graphene fences achieved 43.6% LPG ($\text{C}_3^{\text{P}}\text{-C}_4^{\text{P}}$) selectivity and the low CO production of 2.2% at CO_2 conversion of 46%, while Fe-Co with an intimate contact being anchored on the surface of graphene fences presented 50.1% light olefin ($\text{C}_2^{\text{F}}\text{-C}_4^{\text{F}}$) selectivity at a CO_2 conversion of 55.4%. With the assistance of graphene fences, the synergy

between iron carbides and metallic cobalt could efficiently regulate olefin secondary hydrogenation, achieving a selective switch from light olefins to LPG for CO₂ hydrogenation. The catalytic results are very exciting and if the principle works, it can be a breakthrough in Power to X PTX industry, which is a crucial platform technology for future net zero economy. Various catalysts samples like GO-Fe/K-Co and GO/K-Fe-Co and nanotube supported Co-Fe have been prepared and lots of interesting results are obtained, in general, it is shown that graphene fence plays a big role in the process and the metal loading ways also has big influence. In general it is a nice paper, and can be accepted for publication after the following being addressed:

Responses:

My co-authors and I would like to sincerely thank Referee #3 for his professional review and positive evaluation of our work. The positive opinions of Referee #3 have deeply inspired our further research.

In this work, we first employed a “self-formed graphene fence” to separate the Fe-Co active sites. The different distributions of active sites switched product types between light olefins (C₂⁼-C₄⁼) and LPG (C₃^P-C₄^P) without changing the total metal content and achieved good catalytic performances both in light olefin and LPG production. These results are innovative both in the utilization of graphene support and in the product regulation of CO₂ hydrogenation. Furthermore, this manuscript creates a precedent for CO₂ hydrogenation to LPG via a Fischer-Tropsch pathway and greatly improves the STY (space-time yield) of LPG compared to the previously reported methanol-intermediate composite catalysts, which could better meet the needs of industrial production in the context of increasing demand for carbon-neutral LPG.

We appreciate Referee #3 once again for the assessment and approval, and we will address each of his/her comments and concerns in detail below.

Specific Comments:

Comment 1. The catalyst test section, I cannot find how much catalyst samples were used for the test, and information on the catalyst physical properties. Given the unique performance of the catalysts, physical properties like shape, particle size play key roles in the reactants and products penetration and diffusion.

Responses:

We apologize for missing the specifics of the experiment. We also thank Referee #3 for the careful review, which makes our work more complete.

First, considering the small density of the graphene support, excessive catalyst weight may result in a large catalyst bed height, making it difficult to control the temperature. Therefore we take catalyst of 0.12 g and match it with a flow rate of 10 mL/min reaction gas, so that W/F was controlled at 4.5 g h mol⁻¹.

The catalysts were granulated into granules of 20–40 meshes (0.425 mm–0.85 mm), as most of the past works did (*Nat Commun.* 2022, 13, 2396; *ACS Catal.* 2020, 10, 12098–12108; *The Innovation.* 2023, 4, 100445), to maintain high catalytic performance and diffusion capability. Furthermore, 0.5 g quartz sand (20–40 meshes) was used to physically mix with the catalysts in order to disperse the catalysts and prevent them from sintering. On top of the catalyst bed, 1 g glass beads are applied to adjust the bed height and preheat the reaction gas. The two ends of the catalyst bed and glass beads were separated by quartz cotton. The detailed catalyst dimensions and filling method, together with the catalyst evaluation device information, are shown in **Scheme R7**.

We have added the detailed catalyst filling method to **Pages 35 and 36** in red words in the revised manuscript.

Scheme R7 | Schematic diagram of the catalyst evaluation device and detailed catalyst filling method.

Comment 2. Why $C_{15}H_{21}CoO_6$ was used for Co source? Can $Co(NO_3)_2$ make any difference?

Responses:

We would like to thank Referee #3 for his meticulous observation and consideration of our manuscript. As we already responded to Referee #1, due to the lower target content of Co (4%) compared to that of Fe (20%), we conducted a preliminary screening of varied cobalt sources before the experiment to reduce cobalt loss in hydrothermal synthesis processes and obtain more accurate cobalt content. We found that cobalt acetylacetonate is more easily anchored to graphene than cobalt nitrate, as shown in **Table R5**.

We employed cobalt acetylacetonate and cobalt nitrate containing the same molar amount of Co as varied Co sources for the hydrothermal synthesis with the same weight of graphene oxide (GO). After the same steps of hydrothermal synthesis, filtration, washing, and drying, ICP-OES was applied to the two samples to obtain the Co contents. As shown in **Table R5**, the sample using cobalt acetylacetonate as the Co source exhibited higher Co content. This result could be explained by the affinity between organic groups on graphene surface and cobalt acetylacetonate.

Meanwhile, cobalt-based catalysts with cobalt acetylacetonate as sources have been proven to be well dispersed on graphene (*Catal Today*. 2020, 355, 10–16). As a result, in order

to make the Co content easier to manipulate and to get the well dispersed Co particles, we employed cobalt acetylacetonate as the Co source in the whole process.

Table R5 | Hydrothermal synthesis results of GO with different Co sources.

Co sources	Co molar of Co sources	Weight of Co sources	GO weight	Co content ^a
C ₁₀ H ₁₆ CoO ₄	0.002 mol	0.518 g	2 g	5.43%
Co(NO ₃) ₂ ·6H ₂ O	0.002 mol	0.582 g	2 g	5.21%

^a Co content was obtained by ICP-OES tests.

Comment 3. How about the temperature effect on the product distribution and CO₂ conversion?

This may be relevant to the reaction mechanism.

Responses:

We would like to thank Referee #3 for his valuable suggestions. We tested the GO-Fe/K-Co catalyst at different temperatures, from 280 °C to 340 °C, to illustrate the temperature effect on the catalytic performances. As shown in Table R6, with the temperature increasing, the CO₂ conversion exhibited an increasing trend, while the CO selectivity gradually decreased, which indicated that the increase in temperature is conducive to the activation and hydrogenation of CO₂, as well as pushing formed CO into hydrocarbons, thus promoting the reaction.

As for the product distributions, the CH₄ selectivity increased with the temperature rising. On the contrary, the selectivity of C₂⁼-C₄⁼ and C₅⁺ decreased, demonstrating that excessively high temperatures favored methanation reactions and inhibited C-C coupling reactions (*Fuel*, 2023, 333, 126412). Therefore, considering C-C coupling and olefin hydrogenation capacity comprehensively, LPG selectivity reached the highest value at 320 °C (40.8%).

Table R6 | Catalytic performances of GO-Fe/K-Co catalyst with different reaction temperatures.

Temperature (°C)	CO ₂ conv. (%)	CO sel. (%)	Selectivity (%)					
			CH ₄	C ₂ ⁼ -C ₄ ⁼	C ₂ ^P	C ₃ ^P -C ₄ ^P	C ₄ ^{iso}	C ₅ ⁺
280	33.1	7.1	21.2	8.3	20.9	35.1	4.4	10.1
300	39.6	4.8	23.8	7.1	22.3	35.4	3.9	7.5
320	45.5	2.6	26.4	2.1	23.5	40.8	1.7	5.8
340	47.5	2.5	32.3	1.9	27.2	33.3	1.2	4.1

Reaction conditions: W/F=4.5 g h mol⁻¹ (relative to the total catalyst weight), 3.0 MPa, reduced

in pure hydrogen for 8 hours before reaction.

Comment 4. How about the metal and graphene changes in the catalysts after the activity tests? Ideally the characterization data on the working catalysts at various stage would be great help for understanding the catalyst and reaction.

Responses:

We would like to thank Referee #3 for his valuable comments to improve our manuscript. First, *in situ* XRD was employed to investigate the composition changes of iron in the reduction and reaction processes (**Fig. R15**).

Initially, iron species existed in the form of Fe_2O_3 . With the temperature increasing and the introduction of hydrogen, the peaks gradually shifted in a negative direction, indicating the transformation of Fe_2O_3 to Fe_3O_4 . As the temperature continued to increase, peaks of metallic Fe began to appear. Among them, GO-Fe-Co/K corresponds to the highest temperature of metallic Fe occurring, illustrating the lowest reduction capacity, which is consistent with the H_2 -TPR result (**Supplementary Fig. 19**).

After all the Fe species were reduced to metallic Fe, the atmosphere was switched to reaction gas (CO_2/H_2), and the temperature was lowered to 320 °C to simulate the real experimental conditions. As observed, the peak strength of metallic Fe began to decrease, and peaks attributed to Fe_5C_2 appeared. After the carbonization processes were stabilized, GO-Fe-Co/K exhibited the highest metallic Fe peak and the lowest Fe_5C_2 peak, while GO/K-Fe-Co showed the lowest metallic Fe peak and the highest Fe_5C_2 peak, which corresponded to the Fe_5C_2 content results obtained by the ^{57}Fe Mössbauer spectra (**Fig. 2c** and **Supplementary Table 8**). These results indicated that GO/K-Fe-Co had the best carbonization capacity, while GO-Fe-Co/K exhibited lowest reduction and carbonization abilities due to its strong metal-support interaction (SMSI).

Subsequently, we tested the *in situ* XRD of GO under the hydrogen and reaction gas (CO_2/H_2) atmosphere to explore the transformation of graphene support during the reduction and reaction processes. In the reduction process, with the temperature increasing, the

characteristic peak of graphene (peak 002) at about 8° gradually decreased and shifted in the positive direction. When the temperature reached 205 °C, a new peak appeared near 22° and its peak intensity gradually increased and moved in a positive direction. However, it no longer shifted obviously when moving to about 25° and remained stable, representing the complete reduction of graphene (**Fig. R16a**). According to Bragg's Law (*J. Electron. Spectrosc. 2014, 195, 145–154*), the positive shift of the graphene peak during the reduction is attributed to the gradual decrease of graphene layer spacings, which is consistent with our characterization results.

The gas was subsequently switched to the reaction gas (CO₂/H₂), and the temperature was lowered to 320 °C. Under these conditions, the characteristic peak of graphene did not change significantly (**Fig. R16b**), indicating that graphene would not be oxidized or structurally damaged during the reaction process.

Furthermore, TEM mapping images of spent GO-Fe/K-Co were used to observe the Fe and Co distributions in the spent GO-Fe/K-Co catalyst (**Fig. R17**). Similar to the metal distributions of the as-prepared GO-Fe/K-Co (**Fig. 1f**), there were still some regions with different Fe and Co distributions (obvious Fe distributions but few Co distributions), as labeled by red circles, indicating that the dual active site distributions were not damaged after reduction and reaction. Combined with the fact that particle sizes of GO-Fe/K-Co and graphene layer spacings remained stable during the reaction (**Supplementary Fig. 16 and Fig. R16**), the structure of GO-Fe/K-Co did not change significantly after the reaction, and only the metal phase composition was transformed during the reduction and carbonization processes.

Figs. R15–17 have been added as **Supplementary Fig. 21, Supplementary Fig. 2, and Supplementary Fig. 24** in the revised supporting information file, respectively. And the descriptions were added to **Page 17, Page 7, and Page 20**, respectively, in red words in the revised manuscript.

Fig. R15 | *In situ* XRD patterns for the reduction and carbonization processes of different catalysts. Test conditions: Pure H₂ from 25–400 °C and reaction gas (CO₂/H₂) at 320 °C, atmospheric pressure, 30 mL/min. The left images represent the reduction process in a hydrogen atmosphere, and the right images represent the carbonization process in a reaction gas atmosphere.

Fig. R16 | *In situ* XRD patterns of GO in hydrogen (a) and reaction gas atmosphere (b). Test conditions: Pure H₂ from 25–400 °C and reaction gas (CO₂/H₂) at 320 °C, atmospheric pressure, 30 mL/min.

Fig. R17 | TEM mapping images of spent GO-Fe/K-Co catalyst. The bar stands for 30 nm. The red circles represent the regions where Fe and Co exhibited different distributions.

References (cited in Fig. R2)

1. Li, H. et al. A well-defined core-shell-structured capsule catalyst for direct conversion of CO₂ into liquefied petroleum gas. *ChemSusChem*. **13**, 2060–2065 (2020).
2. Fujiwara, M. et al. Synthesis of C₂₊ hydrocarbons by CO₂ hydrogenation over the composite catalyst of Cu–Zn–Al oxide and HB zeolite using two-stage reactor system under low pressure. *Catal. Today*. **242**, 255–260 (2015).
3. Wang, S. et al. Highly selective hydrogenation of CO₂ to propane over GaZrO_x/H-SSZ-13 composite. *Nat. Catal.* **5**, 1038–1050 (2022).
4. Li, C., Yuan, X. & Fujimoto, K. Direct synthesis of LPG from carbon dioxide over hybrid catalysts comprising modified methanol synthesis catalyst and β-type zeolite. *Appl. Catal. A*. **475**, 155–160 (2014).
5. Natakaranakul, J. et al. Direct synthesis of liquefied petroleum gas from carbon dioxide using a copper/zinc oxide/zirconia/alumina and HY zeolite hybrid catalyst. *ChemistrySelect*. **6**, 7103–7110 (2021).
6. Lu, S. et al. Highly selective synthesis of LPG from CO₂ hydrogenation over In₂O₃/SSZ-13 bifunctional catalyst. *J. Fuel. Chem. Technol.* **49**, 1132–1139 (2021).

REVIEWERS' COMMENTS

Reviewer #1 (Remarks to the Author):

The revised manuscript has been improved and can be recommended for publication in Nature Communications.

Reviewer #2 (Remarks to the Author):

The authors have made important improvements in the manuscript that I believe address most of the referees' comments. I recommend acceptance in its current form.

Reviewer #3 (Remarks to the Author):

The revisions addressed my questions and concerns about the paper, so I am happy to recommend for publication.

Point-by-point responses to the comments (NCOMMS-23-31318A)

We are very grateful to the editor and referees for the review and valuable suggestions on the manuscript. The following descriptions are the responses to these comments, one by one. The questions from the reviewers were highlighted in light blue words.

Reviewer #1 (Remarks to the Author):

Overall Comments: The revised manuscript has been improved and can be recommended for publication in Nature Communications.

Responses:

We are grateful that Referee #1 found the innovation and significance of our work during the review process. We would like to thank Referee #1 for the professional review work, constructive comments, and valuable suggestions on our manuscript.

Reviewer #2 (Remarks to the Author):

Overall comment: The authors have made important improvements in the manuscript that I believe address most of the referees' comments. I recommend acceptance in its current form.

Responses:

We gratefully thank Referee #2 for the timely feedback on our work. We also sincerely thank Referee #2 for the helpful comments and appreciation, which would help to improve the

quality of our manuscript.

Reviewer #3 (Remarks to the Author):

Overall Comment: The revisions addressed my questions and concerns about the paper, so I am happy to recommend for publication.

Responses:

My co-authors and I would like to deeply thank Referee #3 for his professional review and encouraging comments on our work. These constructive comments greatly complemented the details and improved the manuscript. Meanwhile, the positive opinions gave us great motivation for further research on this work.